# A tRNA processing enzyme is a key regulator of the mitochondrial unfolded protein response

**James P Held[1], Gaomin Feng[2], Benjamin R Saunders[1], Claudia V Pereira[1], Kristopher Burkewitz[2], Maulik R Patel[1,2,3]***

[1]Department of Biological Sciences, Vanderbilt University, Nashville, United States; [2]Department of Cell and Developmental Biology, Vanderbilt University, Nashville, United States; [3]Diabetes Research and Training Center, Vanderbilt University School of Medicine, Nashville, United States

**Abstract** The mitochondrial unfolded protein response (UPR^mt) has emerged as a predominant mechanism that preserves mitochondrial function. Consequently, multiple pathways likely exist to modulate UPR^mt. We discovered that the tRNA processing enzyme, homolog of ELAC2 (HOE-1), is key to UPR^mt regulation in *Caenorhabditis elegans*. We find that nuclear HOE-1 is necessary and sufficient to robustly activate UPR^mt. We show that HOE-1 acts via transcription factors ATFS-1 and DVE-1 that are crucial for UPR^mt. Mechanistically, we show that HOE-1 likely mediates its effects via tRNAs, as blocking tRNA export prevents HOE-1-induced UPR^mt. Interestingly, we find that HOE-1 does not act via the integrated stress response, which can be activated by uncharged tRNAs, pointing toward its reliance on a new mechanism. Finally, we show that the subcellular localization of HOE-1 is responsive to mitochondrial stress and is subject to negative regulation via ATFS-1. Together, we have discovered a novel RNA-based cellular pathway that modulates UPR^mt.

*For correspondence:
maulik.r.patel@vanderbilt.edu

**Competing interest:** The authors declare that no competing interests exist.

## Editor's evaluation

This manuscript reports a novel RNA-based cellular pathway that modulates mitochondrial UPR (UPRmt). It advances our understanding of the mitochondrial-to-nuclear communication mediated by a tRNA processing enzyme.

## Introduction

Mitochondria are central to a myriad of cellular processes including energy production, cellular signaling, biogenesis of small molecules, and regulation of cell death via apoptosis (*Nunnari and Suomalainen, 2012*). Mitochondrial dysfunction can lead to metabolic and neurological disorders, cardiovascular disease, and cancers (*Vafai and Mootha, 2012*). To maintain proper mitochondrial function cellular mechanisms have evolved that respond to, and mitigate, mitochondrial stress (*Baker et al., 2012*; *Wang and Chen, 2015*; *Wrobel et al., 2015*; *Munkácsy et al., 2016*; *Tjahjono and Kirienko, 2017*; *Weidberg and Amon, 2018*; *Naresh and Haynes, 2019*; *Fessler et al., 2020*; *Guo et al., 2020*).

One of the predominant mitochondrial stress response mechanisms is the mitochondrial unfolded protein response (UPR^mt). Although first discovered in mammals (*Zhao et al., 2002*), UPR^mt has been best characterized in *Caenorhabditis elegans* (*Naresh and Haynes, 2019*). UPR^mt is primarily characterized by transcriptional upregulation of genes whose products respond to and ameliorate mitochondrial stress (*Yoneda et al., 2004*; *Nargund et al., 2012*).

In *C. elegans*, activation of UPR^mt relies on the transcription factor ATFS-1 that primarily localizes to mitochondria, but under mitochondrial-stress conditions is trafficked to the nucleus where it drives the expression of mitochondrial stress response genes (*Haynes et al., 2010*; *Nargund et al., 2012*; *Nargund et al., 2015*). However, it has become increasingly apparent that UPR^mt is under multiple levels of control: Mitochondrial stress in neurons can activate intestinal UPR^mt non-cell-autonomously via retromer-dependent Wnt signaling (*Durieux et al., 2011*; *Berendzen et al., 2016*; *Zhang et al., 2018*); overexpression of two conserved histone demethylases are independently sufficient to activate UPR^mt (*Merkwirth et al., 2016*); and ATFS-1 is post-translationally modified to facilitate its stability and subsequent UPR^mt activation (*Gao et al., 2019*). Given mitochondrial integration into many diverse cellular signaling and metabolic pathways, there are likely yet-to-be identified pathways regulating UPR^mt.

In conducting a small-scale RNAi screen to interrogate the effects of perturbing mitochondrial RNA processing we discovered that the 3'-tRNA zinc phosphodiesterase, homolog of ELAC2 (HOE-1), is a key regulator of UPR^mt in *C. elegans*. ELAC2 is an essential endonuclease that cleaves 3'-trailer sequences from nascent tRNAs—a necessary step of tRNA maturation—in both nuclei and mitochondria (*Nashimoto et al., 1999*; *Mayer et al., 2000*; *Schiffer et al., 2002*; *Takaku et al., 2003*; *Dubrovsky et al., 2004*; *Brzezniak et al., 2011*; *Sanchez et al., 2011*; *Siira et al., 2018*). ELAC2 has also been reported to cleave other structured RNAs yielding tRNA fragments, small nucleolar RNAs (snoRNAs) and micro RNAs (miRNAs) (*Kruszka et al., 2003*; *Lee et al., 2009*; *Bogerd et al., 2010*; *Siira et al., 2018*). In humans, mutations in ELAC2 are associated with hypertrophic cardiomyopathy (*Haack et al., 2013*; *Shinwari et al., 2017*; *Saoura et al., 2019*) and prostate cancer (*Tavtigian et al., 2001*; *Korver et al., 2003*; *Noda et al., 2006*) while in *C. elegans*, loss of HOE-1 has been shown to compromise fertility (*Smith and Levitan, 2004*).

Surprisingly, we find that it is not the mitochondrial, but rather the nuclear activity of HOE-1 that is required for activation of UPR^mt. Remarkably, compromising nuclear export of HOE-1 is sufficient to specifically and robustly activate UPR^mt. Blocking tRNA export from the nucleus suppresses this HOE-1-dependent UPR^mt induction, suggesting that HOE-1 generates RNA species required in the cytosol to trigger UPR^mt. Finally, we show that HOE-1 nuclear levels are dynamically regulated under conditions of mitochondrial

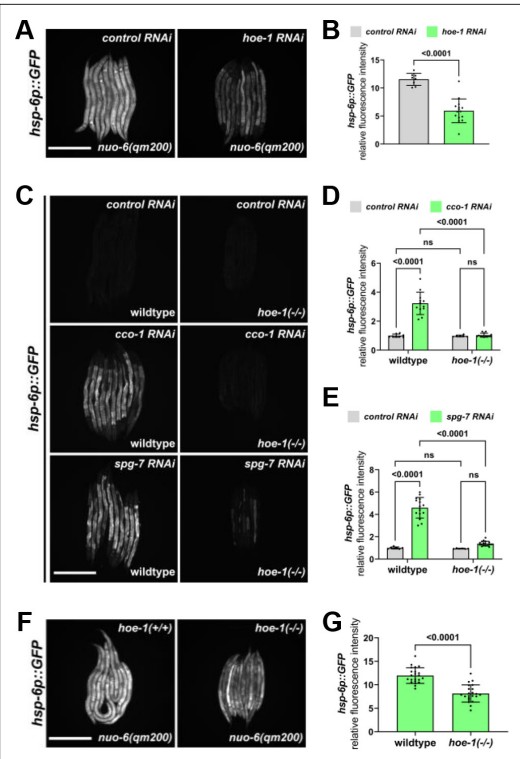

**Figure 1.** *hoe-1* is required for maximal UPR^mt activation. (**A**) Fluorescence images of UPR^mt reporter (*hsp-6p::GFP*) activation in L4 *nuo-6(qm200)* animals on *control* and *hoe-1 RNAi*. Scale bar 200 μm. (**B**) Fluorescence intensity quantification of *hsp-6p::GFP* in individual L4 *nuo-6(qm200)* animals on control and *hoe-1 RNAi* normalized to *hsp-6p::GFP* in a wildtype background on *control RNAi* (n = 8 and 15 respectively, mean and SD shown, unpaired t-test). (**C**) Fluorescence images of UPR^mt reporter (*hsp-6p::GFP*) activation in L3/L4 wildtype and *hoe-1* null (*hoe-1(-/-)*) animals on *control*, *cco-1*, and *spg-7 RNAi*. Scale bar 200 μm. (**D**) Fluorescence intensity quantification of *hsp-6p::GFP* in individual L3/L4 wildtype and *hoe-1(-/-)* animals on *control* and *cco-1 RNAi* (n = 8,12,6 and 13 respectively, mean and SD shown, ordinary two-way ANOVA with Tukey's multiple comparisons test). (**E**) Fluorescence intensity quantification of *hsp-6p::GFP* in individual L3/L4 wildtype and *hoe-1(-/-)* animals on *control* and *spg-7 RNAi* (n = 7,15,6 and 18 respectively, mean and SD shown, ordinary two-way ANOVA with Tukey's multiple comparisons test). (**F**) Fluorescence images of UPR^mt reporter (*hsp-6p::GFP*) activation in L3/L4 *nuo-6(qm200)* animals with (*hoe-1(+/+)*) and without (*hoe-1(-/-)*) *hoe-1*. Scale bar 200 μm. (**G**) Fluorescence intensity quantification of *hsp-6p::GFP* in individual L3/L4 *nuo-6(qm200)* animals with (*hoe-1(+/+)*) and without (*hoe-1(-/-)*) *hoe-1* normalized to *hsp-6p::GFP* in a wildtype background (n = 22 for each condition, mean and SD shown, unpaired t-test).

stress, supporting a physiological role for HOE-1 in mitochondrial stress response. Taken together, our results provide a novel mechanism by which UPRmt is regulated as well as provide critical insight into the biological role of the conserved tRNA processing enzyme, HOE-1.

## Results

### *hoe-1* is required for maximal UPRmt activation

We discovered that RNAi against *hoe-1*, a gene encoding a 3'-tRNA phosphodiesterase, attenuates *hsp-6p::GFP* induction—a fluorescence based transcriptional reporter of UPRmt activation (*Yoneda et al., 2004*). Knockdown of *hoe-1* by RNAi is sufficient to attenuate UPRmt reporter activation induced by a loss-of-function mutation in the mitochondrial electron transport chain (ETC) complex I subunit NUO-6 (*nuo-6(qm200)*) (*Figure 1A and B*).

To further interrogate the potential role of *hoe-1* in UPRmt regulation, we used CRISPR/*Cas9* to generate a *hoe-1* null mutant (*hoe-1(-/-)*) by deleting the open reading frame of *hoe-1* (*Dokshin et al., 2018*). The *hoe-1* null mutants do not develop past late larval stage 3, thus the allele is maintained over a balancer chromosome, *tmC25* (*Dejima et al., 2018*). UPRmt induced by the knockdown of both the mitochondrial protease, *spg-7*, and ETC complex IV subunit, *cco-1*, is robustly attenuated in *hoe-1* null animals (*Figure 1C–E*). Furthermore, UPRmt induced by *nuo-6(qm200)* is attenuated in *hoe-1* null animals similarly to what is seen in *nuo-6(qm200)* animals on *hoe-1* RNAi (*Figure 1F and G*). Taken together, these findings suggest that HOE-1 is generally required for maximal UPRmt activation.

### HOE-1 is dual-targeted to nuclei and mitochondria

To better understand the role of HOE-1 in UPRmt regulation, we sought to identify where HOE-1 functions in the cell. HOE-1 is predicted to localize to both nuclei and mitochondria and this dual-localization has been shown for HOE-1 orthologs in *Drosophila*, mice, and human cell lines (*Dubrovsky et al., 2004*; *Brzezniak et al., 2011*; *Rossmanith, 2011*; *Siira et al., 2018*). To determine the subcellular localization of HOE-1 in *C. elegans*, we C-terminally tagged HOE-1 with GFP at its endogenous locus (HOE-1::GFP). Both *hoe-1::GFP* homozygous and *hoe-1::GFP/hoe-1(-/-)* trans-heterozygous animals grow and develop indistinguishably from wildtype animals suggesting that GFP-tagging HOE-1 does not compromise its essential functions (*Figure 2—figure supplement 1A*). We found that HOE-1 localizes to both mitochondria and nuclei (*Figure 2A*).

### Mitochondrial HOE-1 is not required for UPRmt activation

Given the dual-localization of HOE-1, we questioned whether it is mitochondrial or nuclear HOE-1 that is required for UPRmt activation. To address this question, we created mitochondrial and nuclear compartment-specific loss-of-function mutants of HOE-1 (*Figure 2B*). *hoe-1* contains two functional start codons. Translation beginning from the first start codon (encoding methionine 1 (M1)) produces a protein containing a mitochondrial targeting sequence (MTS). Translation beginning from the second start codon (encoding methionine 74 (M74)), which is 3' to the MTS, produces a nuclear specific protein. This feature is conserved in human ELAC2 and it has been shown that mutating M1 to an alanine produces a mitochondrial-specific knockout (*Brzezniak et al., 2011*). Thus, we used the same approach to create a mitochondrial-specific knockout of *C. elegans* HOE-1 (*hoe-1(ΔMTS)*). This mutation is sufficient to strongly attenuate mitochondrial targeting without impacting nuclear localization (*Figure 2—figure supplement 2A*).

UPRmt reporter activation by *spg-7* and *cco-1* RNAi is not attenuated in *hoe-1(ΔMTS)* animals (*Figure 2C and D* and *Figure 2—figure supplement 3A and B*). In fact, UPRmt reporter activation is slightly elevated in *hoe-1(ΔMTS)* animals relative to wildtype. These data suggest that mitochondrial HOE-1 is not required for UPRmt activation.

### Nuclear HOE-1 is required for UPRmt activation

HOE-1 is predicted to contain two nuclear localization signals (NLS). Given that *hoe-1* null mutant animals are developmentally arrested and *hoe-1(ΔMTS)* animals are superficially wildtype we reasoned that completely ablating nuclear localization of HOE-1 may result in recapitulation of the null phenotype. In effort to disentangle the developmental effects from the effect on UPRmt we ablated only one of the nuclear localization signals of HOE-1. To compromise nuclear localization, we mutated the

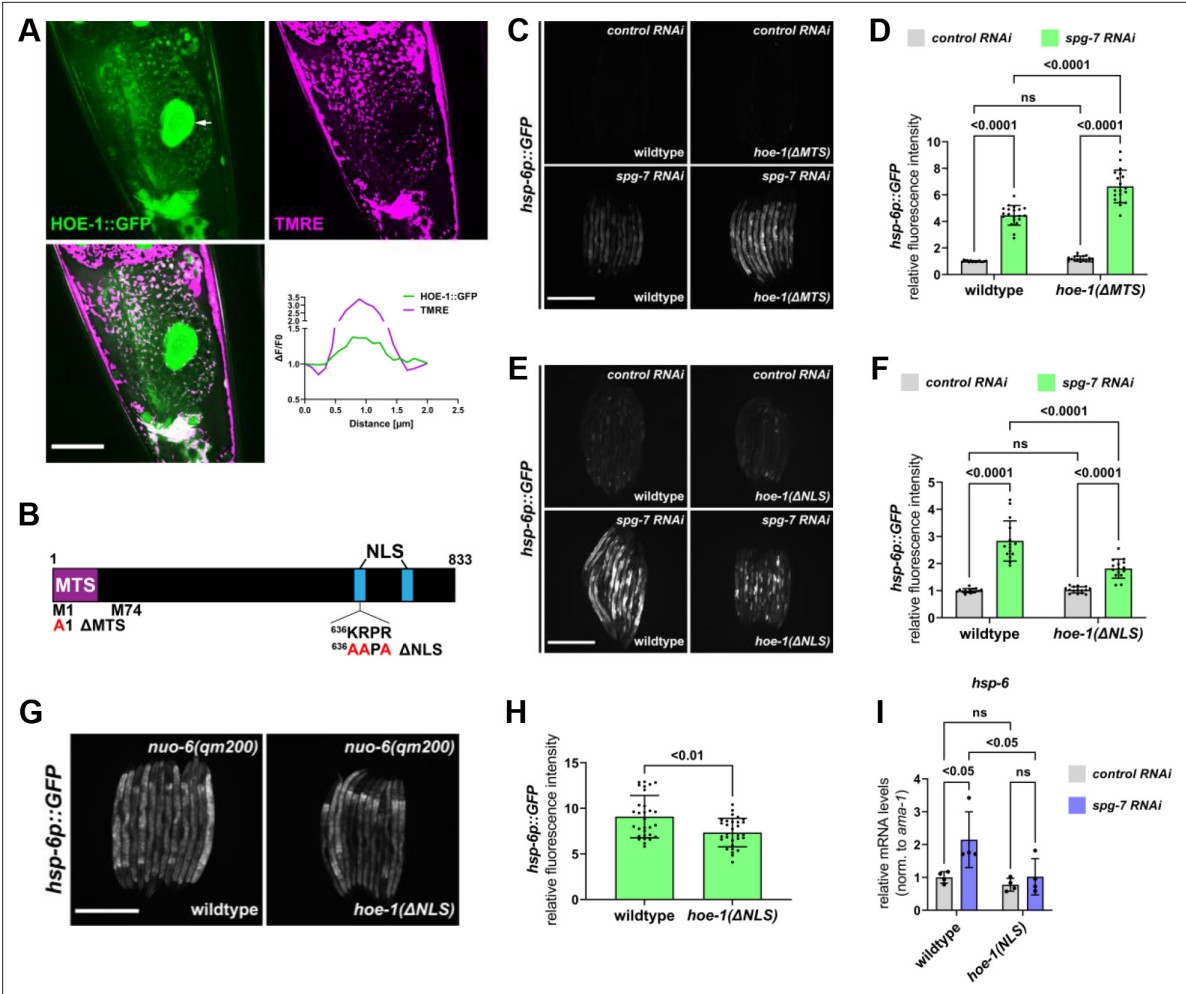

**Figure 2.** Nuclear HOE-1 is required for maximal UPR[mt] activation. (**A**) Fluorescence images of a terminal intestinal cell in a wildtype animal expressing HOE-1::GFP (green) stained with TMRE (magenta) to visualize mitochondria. GFP and TMRE co-localization shown in white in merged image. Arrow indicates nuclei. Scale bar 20 µm. Representative line segment analysis of individual mitochondrion. (**B**) Schematic of HOE-1 protein showing the mitochondrial targeting sequence (MTS) and nuclear localization signals (NLS). ΔMTS allele created by replacing START codon with an alanine (M1A). Transcription begins at M74 for nuclear localized HOE-1. ΔNLS allele created by compromising the most N-terminal NLS ([636]KRPR > AAPA). (**C**) Fluorescence images of UPR[mt] reporter (*hsp-6p::GFP*) in L4 wildtype and *hoe-1(ΔMTS)* animals on *control* and *spg-7 RNAi*. Scale bar 200 µm. (**D**) Fluorescence intensity quantification of *hsp-6p::GFP* in individual L4 wildtype and *hoe-1(ΔMTS)* animals on *control* and *spg-7 RNAi* (n = 15,20,17, and 19 respectively, mean and SD shown, ordinary two-way ANOVA with Tukey's multiple comparisons test). (**E**) Fluorescence images of UPR[mt] reporter (*hsp-6p::GFP*) in L4 wildtype and *hoe-1(ΔNLS)* animals on *control* and *spg-7 RNAi*. Scale bar 200 µm. (**F**) Fluorescence intensity quantification of *hsp-6p::GFP* in individual L4 wildtype and *hoe-1(ΔNLS)* animals on *control* and *spg-7 RNAi* (n = 15 for each condition, mean and SD shown, ordinary two-way ANOVA with Tukey's multiple comparisons test). (**G**) Fluorescence images of UPR[mt] reporter in L4 *nuo-6(qm200)* animals in wildtype and *hoe-1(ΔNLS)* backgrounds. Scale bar 200 µm. (**H**) Fluorescence intensity of *hsp-6p::GFP* in individual L4 *nuo-6(qm200)* animals in wildtype and *hoe-1(ΔNLS)* backgrounds (n = 30 for each condition, mean and SD shown, unpaired t-test). (**I**) mRNA transcript quantification of *hsp-6* in L4 wildtype and *hoe-1(ΔNLS)* animals on *control* and *spg-7 RNAi* normalized to *ama-1* (n = 4 for each condition, mean and SD shown, ordinary two-way ANOVA with Tukey's multiple comparisons test).

The online version of this article includes the following figure supplement(s) for figure 2:

**Figure supplement 1.** *hoe-1::GFP* does not compromise growth or development and is sufficient to rescue the developmental arrest of *hoe-1(-/-)* animals.

**Figure supplement 2.** *hoe-1(ΔMTS)* allele attenuates HOE-1 mitochondrial localization.

**Figure supplement 3.** *hoe-1(ΔMTS)* does not attenuate *cco-1 RNAi*-induced UPR[mt].

**Figure supplement 4.** *hoe-1(ΔNLS)* allele attenuates nuclear HOE-1 localization.

**Figure supplement 5.** UPR[mt]-responsive gene *cyp-14A1.4* is downregulated under mitochondrial stress conditions in *hoe-1(ΔNLS)* animals relative to wildtype.

positively charged residues of the most N-terminal NLS to alanines (*hoe-1(ΔNLS)*). These mutations are sufficient to strongly attenuate, but not completely ablate, HOE-1 nuclear localization whilst still allowing animals to develop to adulthood (*Figure 2—figure supplement 4A–C*).

In contrast to loss of mitochondrial HOE-1, loss of nuclear HOE-1 robustly attenuates UPR^mt reporter activation induced by *spg-7* RNAi (*Figure 2E and F*) and attenuates UPR^mt reporter activation induced by *nuo-6(qm200)* (*Figure 2G and H*). Furthermore, loss of nuclear HOE-1 attenuates the transcriptional upregulation of UPR^mt target genes *hsp-6* (*Figure 2I*) and *cyp-14A1.4* (*Figure 2—figure supplement 5A*) under conditions of mitochondrial stress. Together these data suggest that HOE-1 is required in the nucleus to facilitate UPR^mt activation.

## Compromising HOE-1 nuclear export is sufficient to activate UPR^mt

Like many nuclear localized proteins (*la Cour et al., 2004*), HOE-1 has both nuclear localization signals and a nuclear export signal (NES). Given that loss of nuclear HOE-1 results in UPR^mt attenuation we questioned if compromising HOE-1 nuclear export, by ablating the NES of HOE-1, is sufficient to activate UPR^mt. We created a HOE-1 NES knockout mutant (*hoe-1(ΔNES)*) by replacing the strong hydrophobic residues of the predicted NES with alanines (*Figure 3—figure supplement 1A*). *hoe-1(ΔNES)* animals are superficially wildtype in their development but are sterile. Thus, the allele is balanced with *tmC25*. Homozygous *hoe-1(ΔNES)* animals have elevated nuclear HOE-1 accumulation relative to wildtype (*Figure 3—figure supplement 1B*, *Figure 2—figure supplement 4B and C*).

Strikingly, the UPR^mt reporter *hsp-6p::GFP* is robustly activated in adult *hoe-1(ΔNES)* animals similarly to that seen in mitochondrial stressor *nuo-6(qm200)* and constitutive UPR^mt activation in *atfs-1* gain-of-function (*atfs-1(et15)*) mutant animals (*Figure 3A and B*). *hoe-1(ΔNES)* also mildly induces the less sensitive UPR^mt reporter *hsp-60p::GFP* (*Figure 3—figure supplement 2A and B*).

UPR^mt activation is characterized by the transcriptional upregulation of a suite of mitochondrial stress response genes that encode chaperone proteins, proteases, and detoxification enzymes that function to restore mitochondrial homeostasis (*Nargund et al., 2012*). To interrogate the extent of UPR^mt induction in *hoe-1(ΔNES)* animals, we measured transcript levels of a diverse set of UPR^mt associated genes. We found that the UPR^mt genes encoding a chaperone protein (*hsp-6*), stress response involved C-type lectin (*clec-47*), and P450 enzyme (*cyp-14A4.1*) are all upregulated in *hoe-1(ΔNES)* animals (*Figure 3C, D and E*). These data support *hoe-1(ΔNES)* being sufficient to activate the UPR^mt transcriptional response.

UPR^mt activation is dependent upon the transcription factor ATFS-1 (*Haynes et al., 2010*; *Nargund et al., 2012*). Thus, we tested if UPR^mt reporter activation in *hoe-1(ΔNES)* animals is ATFS-1 dependent. Knockdown of *atfs-1* is sufficient to completely attenuate UPR^mt reporter activation in *hoe-1(ΔNES)* animals (*Figure 3F and G*), showing that UPR^mt induction by *hoe-1(ΔNES)* is ATFS-1 dependent.

## Elevated nuclear HOE-1 levels in *hoe-1(ΔNES)* animals is likely responsible for UPR^mt activation

To further interrogate how UPR^mt is activated in *hoe-1(ΔNES)* animals, we made double localization mutants of *hoe-1*. If UPR^mt is activated in *hoe-1(ΔNES)* animals due to elevated nuclear HOE-1 levels we reasoned that introducing a *hoe-1(ΔNLS)* mutation in the *hoe-1(ΔNES)* background (*hoe-1(ΔNLS+ΔNES)*) should be sufficient to attenuate UPR^mt activation. Indeed, *hoe-1(ΔNLS+ΔNES)* animals have UPR^mt reporter activation comparable to wildtype animals (*Figure 3—figure supplement 3A and B*). Furthermore, we reasoned that compromising mitochondrial localization of HOE-1 in a *hoe-1(ΔNES)* background (*hoe-1(ΔMTS+ΔNES)*) may further enhance *hoe-1(ΔNES)*-induced UPR^mt activation as what would be the mitochondrial targeted HOE-1 pool should be diverted to the nucleus as well. Consistent with our hypothesis, *hoe-1(ΔMTS+ΔNES)* animals have even higher activation of UPR^mt than *hoe-1(ΔNES)* alone (*Figure 3—figure supplement 4A and B*). Taken together, these data strongly suggest that *hoe-1(ΔNES)* triggers UPR^mt activation due to elevated nuclear HOE-1 levels.

## Compromising HOE-1 nuclear export activates UPR^mt cell-autonomously in the intestine

Contrary to UPR^mt induced by *nuo-6(qm200)* and *atfs-1(et15)*, *hoe-1(ΔNES)* animals appear to have UPR^mt activated specifically in the intestine (*Figure 3A*). We questioned if this UPR^mt activation is occurring cell autonomously or non-cell autonomously as UPR^mt has been shown to be able to be

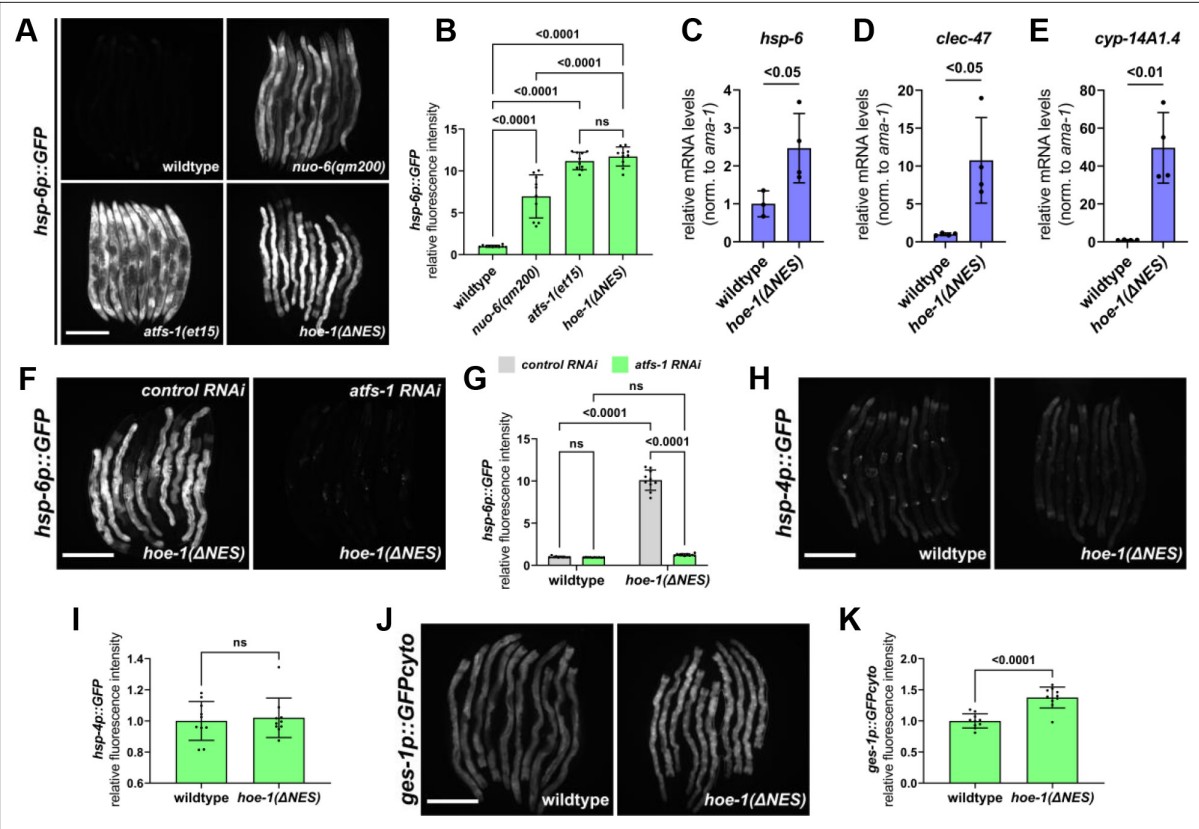

**Figure 3.** Nuclear export defective HOE-1 is sufficient to specifically activate UPR^mt. (**A**) Fluorescence images of UPR^mt reporter (*hsp-6p::GFP*) activation in day 2 adult wildtype, *nuo-6(qm200)*, *atfs-1(et15)*, and *hoe-1(ΔNES)* animals. Scale bar 200 μm. (**B**) Fluorescence intensity quantification of *hsp-6p::GFP* in individual day 2 adult wildtype, *nuo-6(qm200)*, *atfs-1(et15)*, and *hoe-1(ΔNES)* animals (n = 10 for each condition, mean and SD shown, ordinary one-way ANOVA with Tukey's multiple comparisons test). (**C–E**) mRNA transcript quantification of *hsp-6*, *clec-47*, and *cyp-14A1.4*, respectively, in day 2 adult wildtype and *hoe-1(ΔNES)* animals normalized to *ama-1* mRNA levels (n = 4 for each condition, mean and SD shown, unpaired t-test). (**F**) Fluorescence images of UPR^mt reporter (*hsp-6p::GFP*) activation in day 2 adult *hoe-1(ΔNES)* animals on *control* and *atfs-1 RNAi*. Scale bar 200 μm. (**G**) Fluorescence intensity quantification of *hsp-6p::GFP* in individual day 2 adult wildtype and *hoe-1(ΔNES)* animals on *control* and *atfs-1 RNAi* (n = 10 for each condition, mean and SD shown, ordinary two-way ANOVA with Tukey's multiple comparisons test). (**H**) Fluorescence images of UPR^ER reporter (*hsp-4p::GFP*) activation in day 2 adult wildtype and *hoe-1(ΔNES)* animals. Scale bar 200 μm. (**I**) Fluorescence intensity quantification of *hsp-4p::GFP* in individual day 2 adult wildtype and *hoe-1(ΔNES)* animals (n = 10 for each condition, mean and SD shown, unpaired t-test). (**J**) Fluorescence images of intestinal-specific basal protein reporter (*ges-1p::GFPcyto*) activation in day 2 adult wildtype and *hoe-1(ΔNES)* animals. Scale bar 200 μm. (**K**) Fluorescence intensity quantification of *ges-1p::GFPcyto* in individual day 2 adult wildtype and *hoe-1(ΔNES)* animals (n = 10 for each condition, mean and SD shown, unpaired t-test).

The online version of this article includes the following figure supplement(s) for figure 3:

**Figure supplement 1.** Nuclear export defective HOE-1 has increased nuclear accumulation relative to wildtype.

**Figure supplement 2.** Nuclear export defective HOE-1 activates UPR^mt.

**Figure supplement 3.** Compromised nuclear import of HOE-1 completely attenuates *hoe-1(ΔNES)*-induced UPR^mt.

**Figure supplement 4.** Compromised mitochondrial import of HOE-1 exacerbates *hoe-1(ΔNES)*-induced UPR^mt.

**Figure supplement 5.** Nuclear export defective HOE-1 activates UPR^mt in the intestine cell autonomously.

signaled across tissues, particularly from neurons to intestine (*Durieux et al., 2011*; *Berendzen et al., 2016*; *Zhang et al., 2018*). To determine which tissue HOE-1 is required in for UPR^mt activation we took advantage of the auxin-inducible degradation (AID) system that allows for tissue-specific protein degradation (*Zhang et al., 2015*). Briefly, degron-tagged proteins will be degraded in the presence of the plant hormone auxin but only in tissues wherein E3 ubiquitin ligase subunit, TIR1, is expressed. We C-terminally degron-tagged *hoe-1(ΔNES)* (*hoe-1(ΔNES)::degron*) and crossed this allele into backgrounds in which TIR1 is driven under an intestinal-specific (*ges-1p::TIR1*) or a neuronal-specific (*rgef-1p::TIR*) promoter (*Ashley et al., 2021*). *hoe-1(ΔNES)*-induced UPR^mt is only attenuated when

HOE-1 is selectively degraded in the intestine (*Figure 3—figure supplement 5A and B*). This data strongly suggests that compromised nuclear export of HOE-1 activates UPR^mt cell-autonomously in the intestine.

## Compromising HOE-1 nuclear export specifically activates UPR^mt

Changes in protein synthesis rates and associated protein folding capacity can broadly activate cellular stress response mechanisms (*Wang and Kaufman, 2016*; *Das et al., 2017*; *Boos et al., 2019*). Given the role of *hoe-1* in tRNA maturation, we questioned if the robust upregulation of UPR^mt in *hoe-1(ΔNES)* animals may be the result of compromised cellular proteostasis in general rather than specific activation of UPR^mt. One stress response mechanism that is sensitive to global proteotoxic stress is the endoplasmic reticulum unfolded protein response (UPR^ER) (*Preissler and Ron, 2019*). We find that the UPR^ER reporter *hsp-4p::GFP* is not activated in *hoe-1(ΔNES)* animals (*Figure 3H and I*), suggesting that *hoe-1(ΔNES)* does not cause ER stress nor cellular proteotoxic stress. Additionally, a basal reporter of GFP that has been used to proxy general protein expression (*Gitschlag et al., 2016*), *ges-1p::GFPcyto*, is only mildly upregulated in *hoe-1(ΔNES)* animals relative to wildtype (*Figure 3J and K*). Together these findings support that impaired nuclear export of HOE-1 specifically activates UPR^mt.

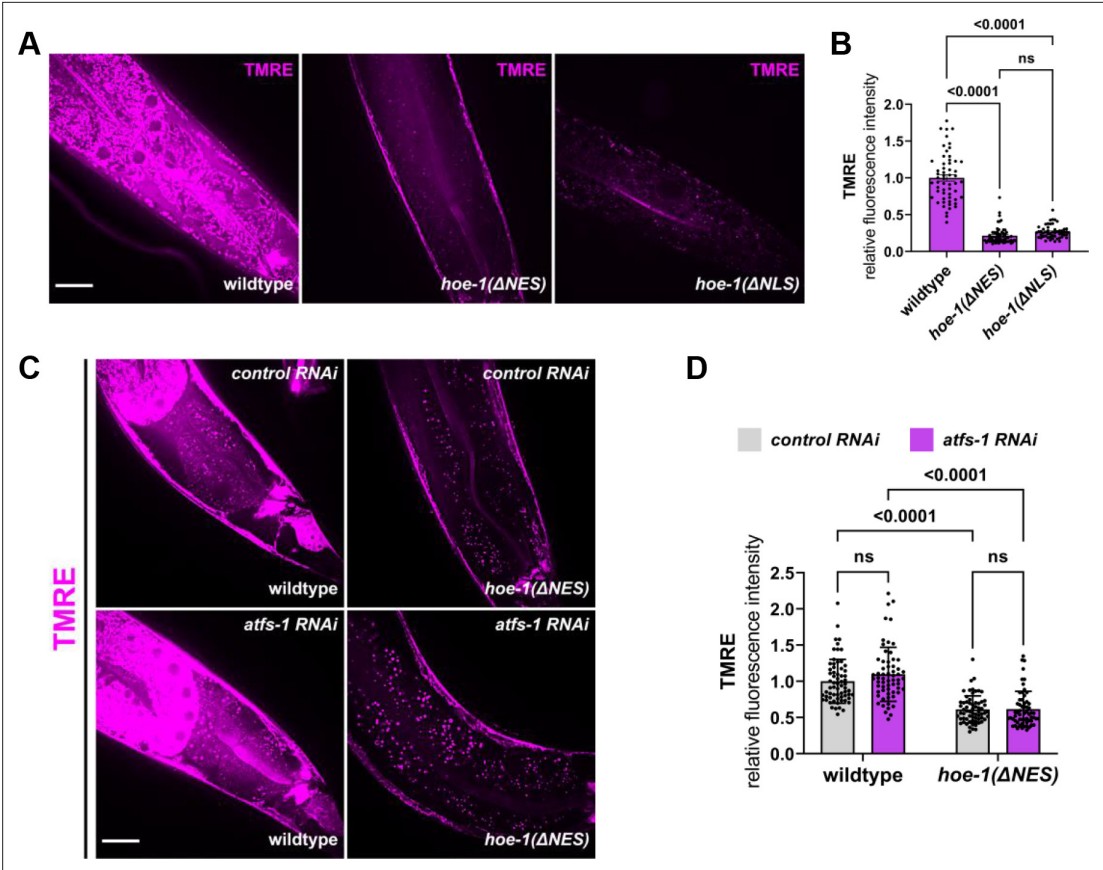

**Figure 4.** Nuclear export defective HOE-1 activates UPR^mt, correlating with reduced mitochondrial membrane potential. (**A**) Fluorescence images of TMRE stained day 1 adult wildtype, *hoe-1(ΔNES)*, and *hoe-1(ΔNLS)* individuals. Scale bar 20 µm. (**B**) Fluorescence intensity quantification of TMRE staining in individual day 1 adult wildtype, *hoe-1(ΔNES)*, and *hoe-1(ΔNLS)* animals (n = 57, 60, and 63 respectively, mean and SD shown, ordinary one-way ANOVA with Tukey's multiple comparisons test). (**C**) Fluorescence images of TMRE stained day 1 adult wildtype and *hoe-1(ΔNES)* animals on *control* and *atfs-1 RNAi*. Scale bar 20 µm. (**D**) Fluorescence intensity quantification of TMRE staining in individual day 1 adult wildtype and *hoe-1(ΔNES)* animals on *control* and *atfs-1* RNAi (n = 65, 62, 65, and 61 respectively, mean and SD shown, ordinary two-way ANOVA with Tukey's multiple comparisons test).

## Compromising HOE-1 nuclear export reduces mitochondrial membrane potential

UPR^mt is known to be triggered when mitochondrial membrane potential is compromised (*Rolland et al., 2019*; *Shpilka et al., 2021*). Thus, we assessed mitochondrial membrane potential, using TMRE staining, in adult *hoe-1(ΔNES)* animals where UPR^mt is robustly activated. Consistent with UPR^mt activation, we found that mitochondrial membrane potential is severely depleted in adult *hoe-1(ΔNES)* animals (*Figure 4A and B*). However, *hoe-1(ΔNLS)* animals also exhibit compromised mitochondrial membrane potential without UPR^mt activation suggesting that decreased membrane potential does not guarantee UPR^mt induction (*Figure 4A and B*). Compromised mitochondrial membrane potential can be both a cause and consequence of UPR^mt activation (*Rolland et al., 2019*; *Shpilka et al., 2021*). Thus, we assessed whether or not compromised membrane potential in *hoe-1(ΔNES)* animals is *atfs-1*-dependent. Mitochondrial membrane potential is not rescued in *hoe-1(ΔNES)* animals on *atfs-1* RNAi (*Figure 4C and D*) suggesting that reduced mitochondrial membrane potential in *hoe-1(ΔNES)* animals is not a result of UPR^mt activation. Taken together, these data show that compromised nuclear export of HOE-1 results in depletion of mitochondrial membrane potential. Furthermore, this depletion in membrane potential correlates with UPR^mt activation, consistent with the possibility that *hoe-1(ΔNES)* activates UPR^mt via depletion of mitochondrial membrane potential.

## Compromising HOE-1 nuclear export elevates nuclear levels of UPR^mt transcription factors ATFS-1 and DVE-1

UPR^mt activation is dependent upon nuclear accumulation of the transcription factor ATFS-1 (*Nargund et al., 2012*; *Nargund et al., 2015*). Thus, we tested if ATFS-1 accumulates in nuclei of *hoe-1(ΔNES)* animals by assessing the fluorescence intensity of ectopically expressed mCherry-tagged ATFS-1 (*atfs-1p*::ATFS-1::mCherry) in wildtype, *hoe-1(ΔNES)*, and mitochondrial-stressed *nuo-6(qm200)* animals. Both *hoe-1(ΔNES)* and *nuo-6(qm200)* animals have elevated nuclear accumulation of ATFS-1 relative to wildtype (*Figure 5A and B*). However, while *nuo-6(qm200)* animals exhibit elevated levels of total cellular and extranuclear levels of ATFS-1::mCherry relative to wildtype, *hoe-1(ΔNES)* animals do not (*Figure 5C* and *Figure 5—figure supplement 1A*). We find that *atfs-1* mRNA levels are also elevated in *hoe-1(ΔNES)* animals relative to wildtype comparable to that seen in *nuo-6(qm200)* animals (*Figure 5D*).

The transcription factor DVE-1 is required for full UPR^mt activation (*Haynes et al., 2007*; *Tian et al., 2016*). Thus, we asked if DVE-1::GFP nuclear expression is higher in *hoe-1(ΔNES)* than in wildtype animals. We found that accumulation of DVE-1::GFP in intestinal cell nuclei is significantly higher in *hoe-1(ΔNES)* than in wildtype animals (*Figure 5E and F*). Qualitatively, cellular DVE-1::GFP levels appear mildly elevated in *hoe-1(ΔNES)* animals based on actin (*Figure 5G*, *Figure 5—source data 1*), though the difference in DVE-1::GFP levels is not significant when normalized to total protein (*Figure 5H*). Thus, while we cannot rule out the possibility of a slight increase in the cellular levels of DVE-1, elevation in the nuclear localization of DVE-1 in *hoe-1(ΔNES)* animals is the more robust phenotype. Together, these data suggest that UPR^mt induction in *hoe-1(ΔNES)* animals is a result of increased nuclear accumulation of UPR^mt transcription factors ATFS-1 and DVE-1.

## UPR^mt is activated by altered tRNA processing in animals with compromised HOE-1 nuclear export

The canonical role of HOE-1 is to cleave 3'-trailer sequences from nascent tRNAs (*Nashimoto et al., 1999*; *Mayer et al., 2000*; *Schiffer et al., 2002*; *Takaku et al., 2003*; *Dubrovsky et al., 2004*; *Brzezniak et al., 2011*; *Sanchez et al., 2011*; *Siira et al., 2018*). This enzymatic function is dependent upon zinc binding (*Ma et al., 2017*; *Bienert et al., 2017*). Thus, we queried if UPR^mt activation by *hoe-1(ΔNES)* is dependent upon the catalytic activity of HOE-1. To test this, we generated a catalytically-dead HOE-1 mutant by changing an essential aspartate of the zinc-binding pocket of HOE-1 to alanine in both a wildtype (*hoe-1(D624A)*) and *hoe-1(ΔNES)* (*hoe-1(D624A+ΔNES)*) background. Animals homozygous for D624A recapitulate the growth arrest phenotype of the *hoe-1* null mutant precluding us from assessing the impact of D624A on UPR^mt induction in adult *hoe-1(ΔNES)* animals. To overcome this constraint, we assessed UPR^mt activation in *hoe-1(ΔNES)* versus *hoe-1(ΔNES)/hoe-1(D624A+ΔNES)* trans-heterozygous animals. A single copy of catalytically dead *hoe-1* is sufficient to

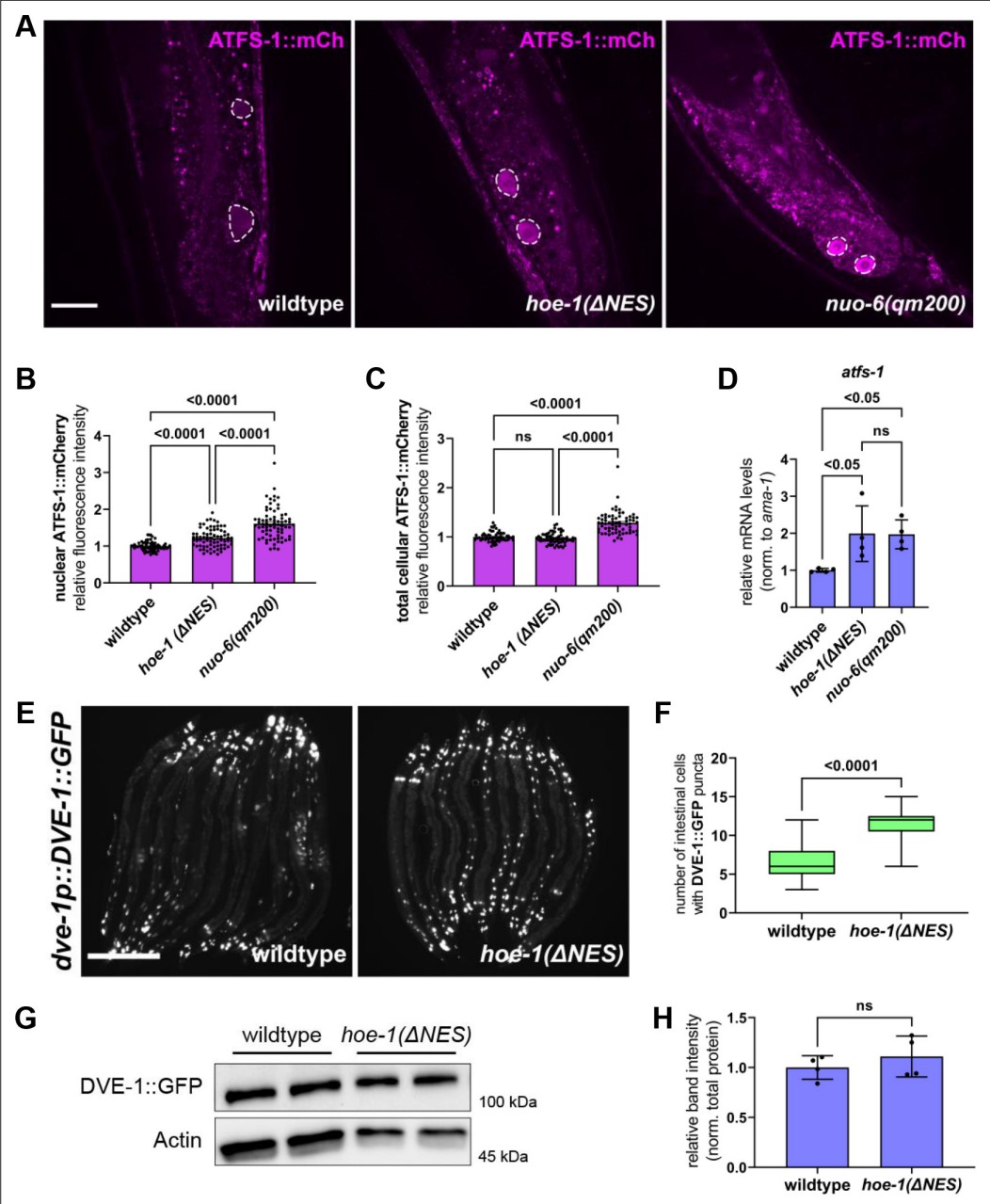

**Figure 5.** Nuclear export defective HOE-1 animals have increased nuclear accumulation of UPR$^{mt}$ transcription factors ATFS-1 and DVE-1. (**A**), Fluorescence images of ATFS-1::mCherry in the terminal intestine of day 2 adult wildtype *hoe-1(ΔNES)*, and *nuo-6(qm200)* individuals (tip of the tail is in the bottom of each panel). Intestinal nuclei outlined with dashed white line. Scale bar 20 μm. (**B**) Fluorescence intensity quantification of nuclear ATFS-1::mCherry in wildtype, *hoe-1(ΔNES)*, and *nuo-6(qm200)* individuals (n = 65, 74, and 72 respectively, mean and SD shown, ordinary one-way ANOVA with Tukey's multiple comparisons test). (**C**) Fluorescence intensity quantification of total cellular ATFS-1::mCherry in wildtype, *hoe-1(ΔNES)*, and *nuo-6(qm200)* individuals (n = 61, 62, and 67 respectively, mean and SD shown, ordinary one-way ANOVA with Tukey's multiple comparisons test). (**D**) mRNA transcript quantification of *atfs-1* in day 2 adult wildtype, *nuo-6(qm200)*, and *hoe-1(ΔNES)* animals normalized to *ama-1* (n = 4 for each condition, mean and SD shown, ordinary one-way ANOVA with Tukey's multiple comparisons test). (**E**) Fluorescence images of *dve-1p::DVE-1::GFP* in day 2 adult wildtype and *hoe-1(ΔNES)* animals. Scale bar 200 μm. (**F**) Number of intestinal cell nuclei with DVE-1::GFP puncta above brightness threshold of 25 in day 2 adult wildtype and *hoe-1(ΔNES)* animals (n = 33 and 41 respectively, unpaired t-test). (**G**) Western blot for DVE-1::GFP and actin from day 1 adult wildtype and *hoe-1(ΔNES)* animals. (**H**) Quantification of DVE-1::GFP western blot band

*Figure 5 continued on next page*

*Figure 5 continued*

intensity from day 1 adult wildtype and *hoe-1(ΔNES)* animals normalized to total protein (n = 4 for each condition, mean and SD shown, unpaired t-test).

The online version of this article includes the following source data and figure supplement(s) for figure 5:

**Source data 1.** Blots for wildtype and *hoe-1(ΔNES)* animals with DVE-1::GFP (*Figure 4G and H*).

**Figure supplement 1.** Nuclear export defective HOE-1 does not elevate extra-nuclear ATFS-1::mCherry levels.

attenuate *hoe-1(ΔNES)*-induced UPR^mt (*Figure 6—figure supplement 1A and B*). These data suggest that *hoe-1(ΔNES)*-induced UPR^mt requires the catalytic activity of HOE-1.

Given that HOE-1 catalytic activity is required for UPR^mt, we further interrogated the potential role of tRNA processing as a mechanism by which HOE-1 may modulate UPR^mt induction. Production of mature tRNAs begins with transcription of tRNA gene loci by RNA polymerase III followed by sequential cleavage of 5'-leader and 3'-trailer sequences from immature tRNA transcripts by RNase P and HOE-1, respectively. Following cleavage of 3'-trailer sequences, tRNAs can be transported to the cytosol by tRNA exportin (*Hopper and Nostramo, 2019*).

Given that HOE-1 nuclear levels are elevated in *hoe-1(ΔNES)* animals, we reasoned that 3'-tRNA processing should be elevated due to increased nuclear activity of HOE-1. Thus, we questioned if UPR^mt induction in *hoe-1(ΔNES)* animals is a result of elevated 3'-tRNA processing. First, we knocked-down RNA pol III subunit *rpc-1* to attenuate the production of total RNA pol III-dependent transcripts in *hoe-1(ΔNES)* animals. If *hoe-1(ΔNES)*-induced UPR^mt is due to elevated processing of tRNAs we hypothesized that restriction of nascent tRNA production should attenuate UPR^mt activation. Indeed, we found that *rpc-1* RNAi robustly attenuates *hoe-1(ΔNES)*-induced UPR^mt (*Figure 6—figure supplement 2A and B*). Interestingly, *rpc-1* RNAi has little impact on mitochondrial stress-induced UPR^mt (*nuo-6(qm200)*) (*Figure 6—figure supplement 2C and D*). These data show that *rpc-1* is required for *hoe-1(ΔNES)*-induced UPR^mt and support our hypothesis that increased 3'-tRNA processing by HOE-1 activates UPR^mt.

For the majority of tRNAs 5'-end processing by the RNase P complex is a prerequisite for 3'-end processing by HOE-1 (*Frendewey et al., 1985*; *Yoo and Wolin, 1997*). Thus, if increased 3'-tRNA end processing is responsible for UPR^mt activation, compromising 5'-end processing by RNAi against RNAse P should attenuate *hoe-1(ΔNES)*-induced UPR^mt. RNAi against a subunit of the RNase P complex, *popl-1*, attenuates UPR^mt induction in *hoe-1(ΔNES)* animals (*Figure 6A and B*). *popl-1* RNAi also attenuates both UPR^mt induced by *nuo-6(qm200)* (*Figure 6C and D*) as well as basal induction of *ges-1p::GFPcyto* (*Figure 6E and F*), albeit to a lesser extent than the attenuation seen in *hoe-1(ΔNES)* animals. These data suggest that *popl-1* RNAi may have a broad impact on protein expression but supports that elevated 3'-tRNA processing in *hoe-1(ΔNES)* animals is responsible for UPR^mt activation given that *popl-1* RNAi strongly attenuates *hoe-1(ΔNES)*-induced UPR^mt.

Following 3'-end processing in the nuclei, tRNAs can be exported to the cytosol by tRNA exportin (*Hopper and Nostramo, 2019*). To test if elevated levels of 3'-processed tRNAs are required in the cytosol to activate UPR^mt, we asked if restricting tRNA nuclear export via RNAi against tRNA exportin, *xpo-3*, attenuates *hoe-1(ΔNES)*-induced UPR^mt. Strikingly, *xpo-3* RNAi robustly attenuates *hoe-1(ΔNES)*-induced UPR^mt (*Figure 6G and H*). However, *xpo-3* RNAi does not attenuate *nuo-6(qm200)* induced UPR^mt (*Figure 6I and J*) nor basal *ges-1p::GFP* levels (*Figure 6K and L*). These data suggest that in *hoe-1(ΔNES)* animals 3'-processed tRNAs are required in the cytosol to activate UPR^mt.

While 5'- and 3'-tRNA processing are the only steps known to be required for tRNA export from the nucleus, there are other downstream tRNA maturation processes that occur (*Hopper and Nostramo, 2019*). Some nascent tRNAs include introns that need to be removed and then ligated by a tRNA ligase (*Englert and Beier, 2005*; *Popow et al., 2012*). For tRNAs to be charged with corresponding amino acids, nascent tRNAs must contain a CCA sequence as part of the 3' acceptor stem. This can be achieved by a CCA-adding tRNA nucleotidyl transferase (*Hou, 2010*). Knockdown of both tRNA ligase, *rtcb-1*, and tRNA nucleotidyl transferase, *hpo-31* mildly attenuate *hoe-1(ΔNES)*-induced UPR^mt (*Figure 6—figure supplement 3A–C*). However, *rtcb-1* RNAi also mildly attenuates *nuo-6(qm200)*-induced UPR^mt (*Figure 6—figure supplement 3D and E*). Knockdown of *hpo-31* severely compromised growth of *nuo-6(qm200)* animals and thus the impact on UPR^mt could not accurately be

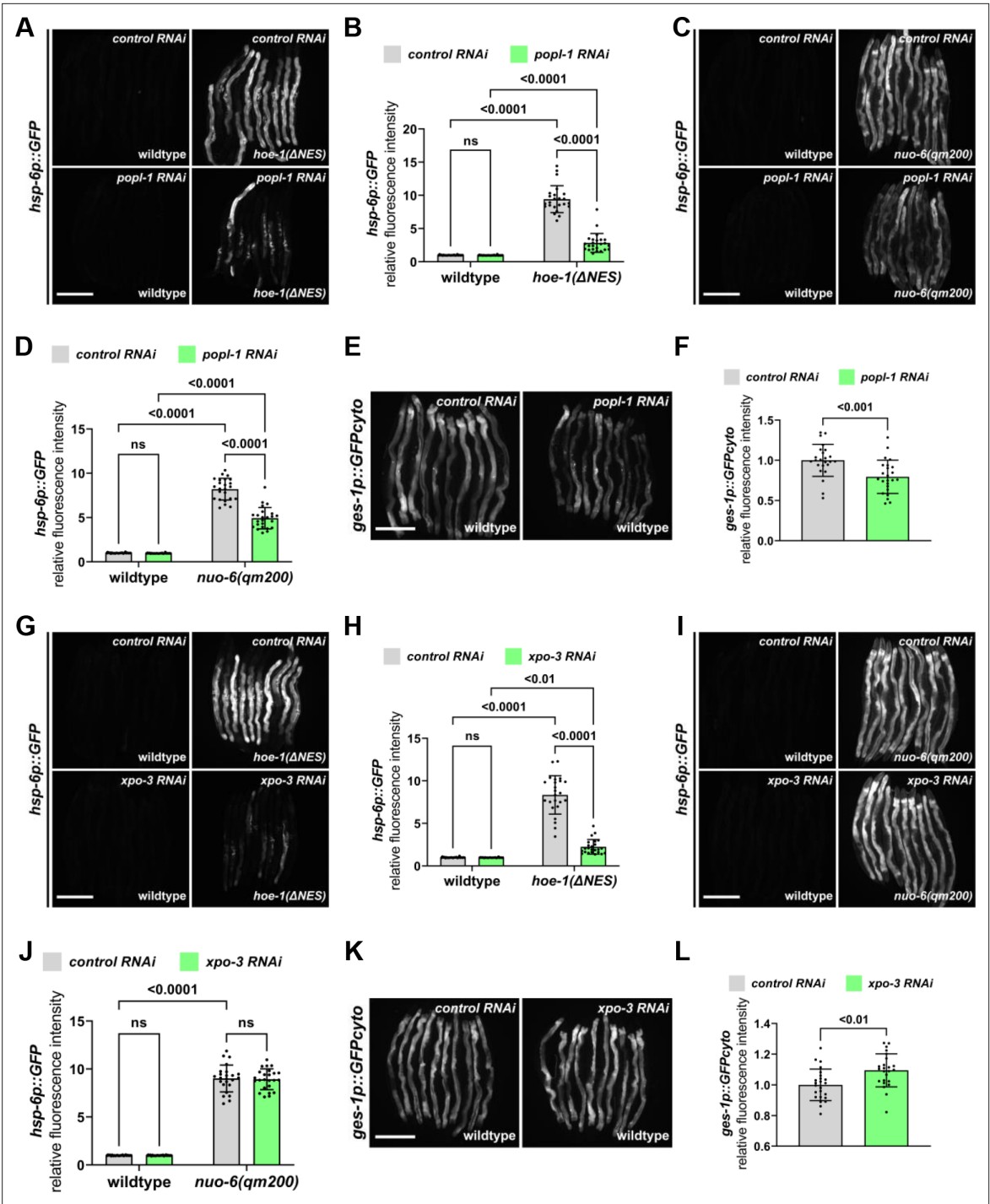

**Figure 6.** Nuclear export defective HOE-1 activates UPR^mt via altered tRNA processing. (**A**) Fluorescence images of UPR^mt reporter (*hsp-6p::GFP*) activation in day 2 adult wildtype and *hoe-1(ΔNES)* animals on *control* and *popl-1 RNAi*. Scale bar 200 µm. (**B**) Fluorescence intensity quantification of *hsp-6p::GFP* in individual day 2 adult wildtype and *hoe-1(ΔNES)* animals on *control* and *popl-1 RNAi* (n = 24 for each condition, mean and SD shown, ordinary two-way ANOVA with Tukey's multiple comparisons test). (**C**) Fluorescence images of UPR^mt reporter (*hsp-6p::GFP*) activation in day 2 adult wildtype and *nuo-6(qm200)* animals on *control* and *popl-1 RNAi*. Scale bar 200 µm. (**D**) Fluorescence intensity quantification of *hsp-6p::GFP* in individual day 2 adult wildtype and *nuo-6(qm200)* animals on *control* and *popl-1 RNAi* (n = 24 for each condition, mean and SD shown, ordinary two-way ANOVA with Tukey's multiple comparisons test). (**E**) Fluorescence images of intestinal-specific basal protein reporter (*ges-1p::GFPcyto*) activation in day 2 adult wildtype animals on *control* and *popl-1 RNAi*. Scale bar 200 µm. (**F**) Fluorescence intensity quantification of *ges-1p::GFPcyto* in individual day 2 adult wildtype animals on *control* and *popl-1 RNAi* (n = 24 for each condition, mean and SD shown, unpaired t-test). (**G**) Fluorescence images of UPR^mt reporter (*hsp-6p::GFP*) activation in day 2 adult wildtype and *hoe-1(ΔNES)* animals on *control* and *xpo-3 RNAi*. Scale bar 200 µm. (**H**) Fluorescence

*Figure 6 continued on next page*

*Figure 6 continued*

intensity quantification of *hsp-6p::GFP* in individual day 2 adult wildtype and *hoe-1(ΔNES)* animals on *control* and *xpo-3 RNAi* (n = 24 for each condition, mean and SD shown, ordinary two-way ANOVA with Tukey's multiple comparisons test). (**I**) Fluorescence images of UPR^mt reporter (*hsp-6p::GFP*) activation in day 2 adult wildtype and *nuo-6(qm200)* animals on *control* and *xpo-3 RNAi*. Scale bar 200 μm. (**J**) Fluorescence intensity quantification of *hsp-6p::GFP* in individual day 2 adult wildtype and *nuo-6(qm200)* animals on *control* and *xpo-3 RNAi* (n = 24 for each condition, mean and SD shown, ordinary two-way ANOVA with Tukey's multiple comparisons test). (**K**) Fluorescence images of intestinal-specific basal protein reporter (*ges-1p::GFPcyto*) activation in day 2 adult wildtype animals on *control* and *xpo-3 RNAi*. Scale bar 200 μm. (**L**) Fluorescence intensity quantification of *ges-1p::GFPcyto* in individual day 2 adult wildtype animals on *control* and *xpo-3 RNAi* (n = 24 for each condition, mean and SD shown, unpaired t-test).

The online version of this article includes the following figure supplement(s) for figure 6:

**Figure supplement 1.** Nuclear export defective HOE-1 induced UPR^mt is dependent upon the catalytic activity of HOE-1.

**Figure supplement 2.** RNAi against RNA polymerase III subunit, *rpc-1*, preferentially attenuates *hoe-1(ΔNES)*-induced UPR^mt.

**Figure supplement 3.** RNAi against tRNA nucleotidyl transferase, *hpo-31*, and tRNA ligase, *rtcb-1*, mildly attenuate both *hoe-1(ΔNES)*- and *nuo-6(qm200)*-induced UPR^mt.

assessed. These data suggest that tRNA ligation and CCA addition have limited involvement in *hoe-1(ΔNES)*-induced UPR^mt.

Taken together, these data suggest that UPR^mt induction by nuclear export deficient HOE-1 is the result of increased 3'-tRNA processing and that these tRNA species are required in the cytosol to trigger UPR^mt.

## Compromised HOE-1 nuclear export does not activate UPR^mt via GCN2 or eIF2α

Alteration to tRNA processing can activate cellular signaling pathways (*Raina and Ibba, 2014*). One such pathway is the integrated stress response (ISR) in which uncharged tRNAs activate the kinase GCN2 which, in turn, phosphorylates the eukaryotic translation initiation factor, eIF2α. This inhibitory phosphorylation of eIF2α leads to upregulation of a select number of proteins including the transcription factor ATF4 (*Pakos-Zebrucka et al., 2016*; *Costa-Mattioli and Walter, 2020*). Interestingly, ATF4

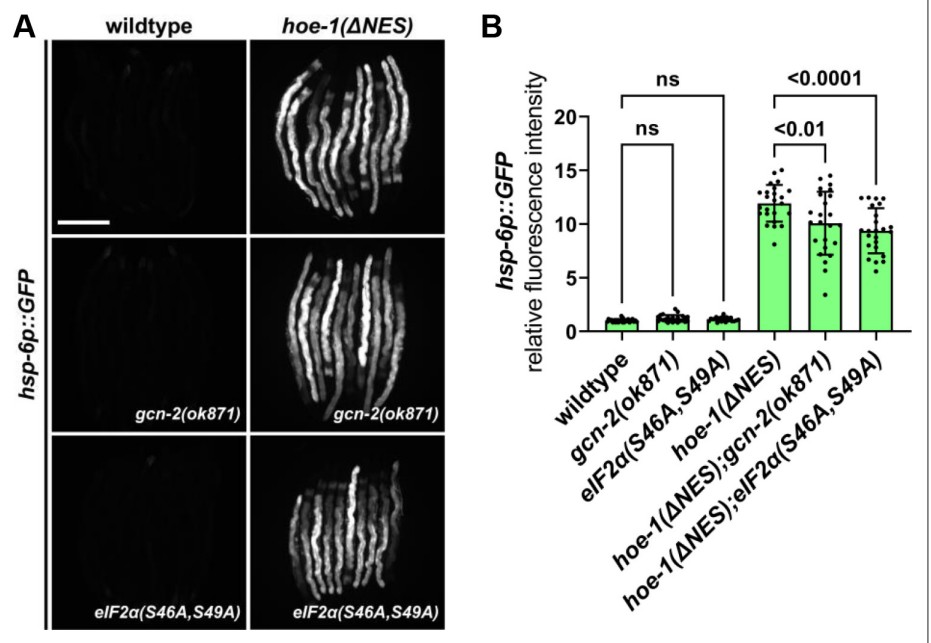

**Figure 7.** Nuclear export defective HOE-1 induced UPR^mt is not *gcn-2* or *eIF2α* dependent. (**A**) Fluorescence images of UPR^mt reporter (*hsp-6p::GFP*) activation in day 2 adult wildtype, *gcn-2(ok871)*, *eIF2α(S46A,S49A)*, *hoe-1(ΔNES)*, *hoe-1(ΔNES);gcn-2(ok871)*, and *hoe-1(ΔNES);eIF2α(S46A,S49A)* animals. Scale bar 200 μm. (**B**) Fluorescence intensity quantification of *hsp-6p::GFP* in individual day 2 adult wildtype, *gcn-2(ok871)*, *eIF2α(S46A,S49A)*, *hoe-1(ΔNES)*, *hoe-1(ΔNES);gcn-2(ok871)*, and *hoe-1(ΔNES);eIF2α(S46A,S49A)* animals (n = 24 for each condition, mean and SD shown, ordinary two-way ANOVA with Tukey's multiple comparisons test).

and one of its targets, ATF5, are orthologs of ATFS-1 (*Fiorese et al., 2016*). Moreover, GCN2 and ISR in general have been shown to be responsive to mitochondrial stress (*Baker et al., 2012*; *Fessler et al., 2020*; *Guo et al., 2020*; *Koncha et al., 2021*). Thus, we questioned if UPR^mt activation by *hoe-1(ΔNES)* is mediated via GCN2 and eIF2α phosphorylation. We found that *hoe-1(ΔNES)*-induced UPR^mt is only slightly reduced in both a *gcn-2* null (*gcn-2(ok871)*) and an *eIF2α* non-phosphorylatable mutant (*eIF2α(S46A,S49A)*) background (*Figure 7A and B*). These data suggest that a mechanism independent of ISR must largely be responsible for UPR^mt activation by *hoe-1(ΔNES)* animals.

## Nuclear HOE-1 is dynamically responsive to mitochondrial stress and negatively regulated by ATFS-1

To better understand the potential physiological implications of HOE-1 in UPR^mt, we assessed *hoe-1* expression and subcellular dynamics of HOE-1 during mitochondrial stress. It is predicted that two major transcripts are produced from the *hoe-1* gene locus: one that contains an MTS and one that does not, which are translated into mitochondrial- and nuclear-targeted HOE-1, respectively. However, it has been shown in other systems that *hoe-1* orthologs produce a single transcript that encodes both a mitochondrial targeted and nuclear targeted HOE-1 isoform via alternative translation initiation (*Rossmanith, 2011*). Thus, we first sought to determine which mechanism is used for *hoe-1* expression. To do so, we designed two sets of primers complementary to *hoe-1* mRNA one of which amplifies only mRNA that includes the MTS and the other which amplifies all *hoe-1* mRNA (spans a sequence that is included in all predicted HOE-1 isoforms). If there are two independent *hoe-1* transcripts, we expected there to be higher levels of *hoe-1* mRNA measured by the primer pair for total transcripts than for the mitochondrial specific pair. However, we found that both primer pairs measured similar levels of *hoe-1* mRNA (*Figure 8—figure supplement 1A*) suggesting that, like in other systems, there is a single *hoe-1* transcript. Next, we assessed *hoe-1* mRNA levels in non-stress versus mitochondrial stress conditions. We found, using both primer pairs, that *hoe-1* mRNA levels are modestly elevated in *nuo-6(qm200)* animals relative to wildtype (*Figure 8—figure supplement 1B and C*) suggesting that *hoe-1* may be transcriptionally upregulated under conditions of mitochondrial stress.

Next, we assessed the subcellular dynamics of HOE-1 in response to mitochondrial stress. We found that HOE-1::GFP nuclear levels are markedly diminished under mitochondrial stress induced by *nuo-6(qm200)*, *cco-1* RNAi, and *spg-7* RNAi (*Figure 8A and B* and *Figure 8—figure supplement 2A and B*). This observation was unexpected given that *hoe-1* transcript levels are elevated during mitochondrial stress and it runs contrary to the fact that compromising HOE-1 nuclear export is sufficient to induce UPR^mt (*Figure 3A and B*). A common feature of signaling pathways is negative regulation. Thus, we questioned if reduced nuclear HOE-1 is a result of negative feedback rather than a direct result of mitochondrial stress. Given that mitochondrial stress activates UPR^mt, we assessed HOE-1::GFP status in a mitochondrial stress background wherein *atfs-1* is knocked down by RNAi. HOE-1 levels are significantly upregulated in nuclei of *nuo-6(qm200)* animals on *atfs-1* RNAi relative to *nuo-6(qm200)* animals on *control* RNAi, as well as both wildtype animals on *control* and *atfs-1* RNAi (*Figure 8A and B* and *Figure 8—figure supplement 3A–C*). Moreover, total cellular HOE-1 levels are elevated under mitochondrial stress in an *atfs-1* RNAi background (*Figure 8C and D* and *Figure 8—source data 1A–E* and *Figure 8—source data 2A–E*). Additionally, mitochondrial HOE-1 levels are elevated under mitochondrial stress conditions irrespective of RNAi treatment (*Figure 8—figure supplement 3D*). Together these data suggest that HOE-1 is upregulated and accumulates in nuclei upon mitochondrial stress. Then, nuclear HOE-1 is negatively regulated by ATFS-1 once UPR^mt is activated.

To further test if nuclear HOE-1 is negatively regulated by UPR^mt activation rather than by mitochondrial stress, we assessed HOE-1 localization in ATFS-1 gain-of-function animals (*atfs-1(et15)*). *atfs-1(et15)* constitutively activates UPR^mt in the absence of mitochondrial stress (*Rauthan et al., 2013*). Thus, we asked if *atfs-1(et15)* is sufficient to reduce nuclear HOE-1 levels. Indeed, nuclear HOE-1 levels are markedly reduced in *atfs-1(et15)* animals relative to wildtype (*Figure 8E and F* and *Figure 8—figure supplement 4A–C*) while total and mitochondrial HOE-1 protein levels are largely unperturbed (*Figure 8G and H*, *Figure 8—figure supplement 4D*, *Figure 8—source data 3A–E* and *Figure 8—source data 4A–E*). These data further support that UPR^mt activation negatively regulates nuclear HOE-1.

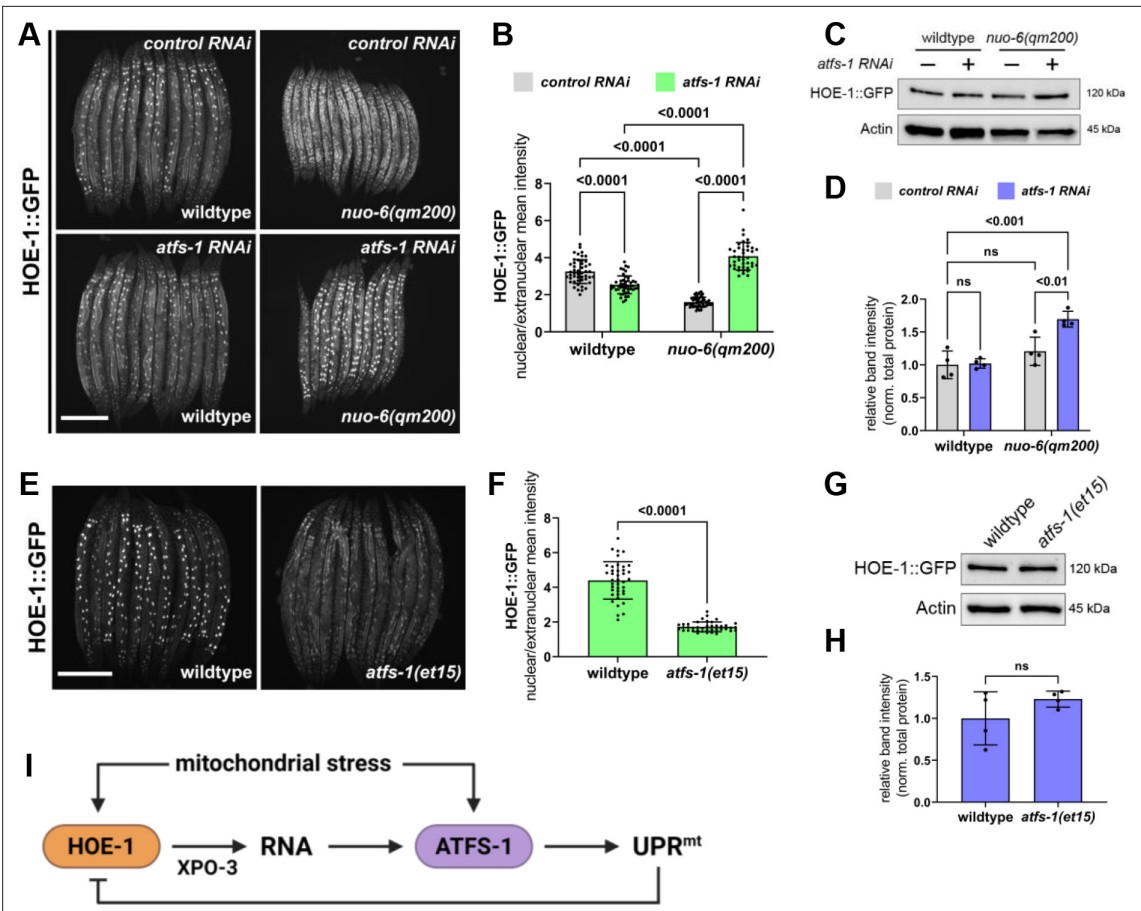

**Figure 8.** Nuclear HOE-1 levels are elevated during mitochondrial stress in the absence of ATFS-1 but decreased in the presence of ATFS-1.
(**A**) Fluorescence images of HOE-1::GFP in day 1 adult wildtype and *nuo-6(qm200)* animals on *control* and *atfs-1 RNAi*. Scale bar 200 μm.
(**B**) Fluorescence intensity quantification of intestinal nuclei relative to extranuclear signal in day 1 adult wildtype and *nuo-6(qm200)* animals on *control* and *atfs-1 RNAi* (n = 40 for each condition, mean and SD shown, ordinary two-way ANOVA with Tukey's multiple comparisons test). (**C**) Western blot for HOE-1::GFP and actin from day 1 adult wildtype and *nuo-6(qm200)* animals on *control* and *atfs-1 RNAi*. (**D**) Quantification of HOE-1::GFP western blot band intensity from day 1 adult wildtype and *nuo-6(qm200)* animals on *control* and *atfs-1 RNAi* normalized to total protein (n = 4 for each condition, mean and SD shown, ordinary two-way ANOVA with Tukey's multiple comparisons test). (**E**) Fluorescence images of HOE-1::GFP in day 1 adult wildtype and *atfs-1(et15)* animals. Scale bar 200 μm. (**F**) Fluorescence intensity quantification of intestinal nuclei relative to extranuclear signal in day 1 adult wildtype and *atfs-1(et15)* animals (n = 40 for each condition, mean and SD shown, unpaired t-test). (**G**) Western blot for HOE-1::GFP and actin from day 1 adult wildtype and *atfs-1(et15)* animals. (**H**) Quantification of HOE-1::GFP western blot band intensity from day 1 adult wildtype and *atfs-1(et15)* animals normalized to total protein (n = 4 for each condition, mean and SD shown, unpaired t-test). (**I**) Mitochondrial stress triggers activation of HOE-1 resulting in altered RNA processing that facilitates UPR^mt via ATFS-1. Activation of UPR^mt negatively regulates HOE-1.

The online version of this article includes the following source data and figure supplement(s) for figure 8:

**Source data 1.** Blots for wildtype and *nuo-6(qm200)* animals on *control* and *atfs-1 RNAi* (***Figure 8C***).

**Source data 2.** Blots for wildtype and *nuo-6(qm200)* animals on *control* and *atfs-1 RNAi* (***Figure 8D***).

**Source data 3.** Blots for wildtype and *atfs-1(et15)* animals (***Figure 8G***).

**Source data 4.** Blots for wildtype and *atfs-1(et15)* animals (***Figure 8H***).

**Figure supplement 1.** *hoe-1* mRNA levels are upregulated under conditions of mitochondrial stress.

**Figure supplement 2.** UPR^mt-inducing *cco-1 and spg-7 RNAi* both attenuate HOE-1 nuclear levels.

**Figure supplement 3.** Nuclear HOE-1 levels are elevated during mitochondrial stress in the absence of ATFS-1 but decreased in the presence of ATFS-1.

**Figure supplement 4.** Constitutive activation of UPR^mt by *atfs-1* gain-of-function (*atfs-1(et15)*) depletes nuclear HOE-1 levels.

## Discussion

Regulation of UPR^mt is not completely understood and elucidating this mechanism has broad implications for understanding cellular response to mitochondrial dysfunction. Here, we describe a novel mechanism by which mitochondrial stress is transduced to activate UPR^mt and how that response is regulated through a feedback mechanism (**Figure 8I**).

Multiple factors have been identified that are required for maximal activation of UPR^mt. This includes the mitochondrial localized proteins, CLPP-1 protease and peptide transmembrane transporter HAF-1 (**Haynes et al., 2007**; **Haynes et al., 2010**). Additionally, the transcription factors ATFS-1 and DVE-1 along with the co-transcriptional activator UBL-5 are required for UPR^mt activation (**Benedetti et al., 2006**; **Haynes et al., 2007**; **Haynes et al., 2010**; **Nargund et al., 2012**; **Nargund et al., 2015**; **Tian et al., 2016**). Histone modifications, chromatin remodeling, and post-translational modifications of ATFS-1 are also involved in fully activating UPR^mt (**Tian et al., 2016**; **Merkwirth et al., 2016**; **Gao et al., 2019**; **Shao et al., 2020**). We show for the first time that nuclear HOE-1 is required for maximal activation of UPR^mt as its induction by various stressors is attenuated in *hoe-1* RNAi, *hoe-1* null, and *hoe-1(ΔNLS)* backgrounds.

We show that loss of *hoe-1* results in varied attenuation of UPR^mt depending on how UPR^mt is activated. UPR^mt induction by RNAi (*cco-1* and *spg-7*) is robustly attenuated by loss of *hoe-1* while *nuo-6(qm200)*-induced UPR^mt is only modestly attenuated. RNAi by feeding works well in all tissues except neurons (**Timmons et al., 2001**; **Kamath et al., 2003**). Importantly, UPR^mt can be activated non-cell autonomously in the intestine by mitochondrial stress in neurons (**Durieux et al., 2011**; **Berendzen et al., 2016**; **Zhang et al., 2018**). UPR^mt induced cell-autonomously in the intestine by RNAi may be *hoe-1* dependent while neuron-to-intestine UPR^mt induction may work primarily in a *hoe-1*-independent manner. Consistent with this, increased nuclear accumulation of HOE-1 only activates UPR^mt in the intestine. These results further exemplify the complexity of UPR^mt signaling.

UPR^mt is generally triggered via compromised mitochondrial membrane potential which facilitates the nuclear accumulation of ATFS-1 (**Rolland et al., 2019**; **Shpilka et al., 2021**). We find that UPR^mt activation via *hoe-1(ΔNES)* correlates with a decrease in mitochondrial membrane potential providing a potential trigger for UPR^mt induction. Furthermore, we show that the UPR^mt transcription factors ATFS-1 and DVE-1 have increased nuclear localization in *hoe-1(ΔNES)* animals, thus likely facilitating the robust UPR^mt activation.

HOE-1 functions in tRNA processing (**Nashimoto et al., 1999**; **Mayer et al., 2000**; **Schiffer et al., 2002**; **Takaku et al., 2003**; **Dubrovsky et al., 2004**; **Brzezniak et al., 2011**; **Sanchez et al., 2011**; **Siira et al., 2018**). Here, we show that increased 3'-tRNA processing by HOE-1 is likely responsible for UPR^mt activation. Restricting HOE-1-dependent 3'-tRNA trailer sequence cleavage indirectly by RNAi against RNA polymerase III subunit, *rpc-1*, and RNase P subunit, *popl-1*, strongly attenuate *hoe-1(ΔNES)*-induced UPR^mt. Moreover, these RNA species must be required in the cytosol to activate UPR^mt as RNAi against tRNA exportin *xpo-3* is sufficient to robustly attenuate *hoe-1(ΔNES)*-induced UPR^mt. Our findings herein are the first reported connection between altered tRNA processing and UPR^mt in *C. elegans*. Given the general requirement for tRNAs in protein translation on the one hand, and the mitochondria-specific nature of UPR^mt on the other, our findings of a connection between the two are intriguing. However, besides performing their core housekeeping function in protein translation, tRNAs have also emerged as small RNAs with important regulatory roles inside cells (**Avcilar-Kucukgoze and Kashina, 2020**). Perhaps, the most well-characterized regulatory role for tRNAs is in the activation of the integrated stress response (ISR). In ISR, uncharged tRNAs activate the eIF2α kinase, GCN2, resulting in the upregulation of ATFS-1 orthologs ATF4 and ATF5 (**Pakos-Zebrucka et al., 2016**; **Costa-Mattioli and Walter, 2020**). However, we show that *gcn-2* and eIF2α are not required for *hoe-1(ΔNES)*-induced UPR^mt activation suggesting that a different mechanism is responsible. The lack of involvement of ISR in HOE-1's role in UPR^mt is not too surprising as there may be a greater pool of fully mature tRNAs in the cytosol in *hoe-1(ΔNES)* animals due to increased 3'-end processing of tRNAs above wildtype levels. This would result in an excess of charged tRNAs in the cytosol, the opposite of what is required to trigger GCN2-dependent ISR. Instead, we can speculate on several additional possibilities for the consequences of increased levels of charged tRNAs that can explain the role of HOE-1 in UPR^mt regulation. For example, the use of amino acids to charge excess tRNAs in *hoe-1(ΔNES)* animals may limit the pool of free amino acids available for mitochondrial import, thus affecting translation of proteins encoded by the mitochondrial genome. This may result

in stoichiometric imbalance between nuclear and mitochondrial-encoded components of the electron transport chain, which is known to compromise mitochondrial membrane potential and trigger UPR$^{mt}$ (*Houtkooper et al., 2013*). Alternatively, mito-nuclear imbalance in *hoe-1(ΔNES)* animals may result from excessive translation of nuclear-encoded mitochondrial proteins due to increased abundance of available charged tRNAs in the cytosol. In yet another scenario, UPR$^{mt}$ may not be the consequence of a global increase in the levels of all cytosolic tRNAs but rather, may be due to changes in the levels of specific tRNAs that preferentially impact translation of genes enriched for the corresponding codons. Such selective upregulation of tRNAs has been shown previously to have specific cellular consequences (*Gingold et al., 2014*; *Goodarzi et al., 2016*). Finally, it is possible that a tRNA-like RNA or other small RNA species such as tRNA fragments are responsible for UPR$^{mt}$ induction in *hoe-1(ΔNES)* animals (*Kruszka et al., 2003*; *Lee et al., 2009*; *Bogerd et al., 2010*; *Siira et al., 2018*). However, if this is the case, our data argue that such an RNA species would need to be transported to the cytosol by tRNA exportin. Non-tRNA transport by an ortholog of *xpo-3* has not yet been reported (*Hopper and Nostramo, 2019*).

We show that nuclear HOE-1 is dynamically regulated by mitochondrial stress. In the presence of stress, nuclear HOE-1 levels are depleted. However, this is UPR$^{mt}$ dependent as HOE-1 nuclear levels under mitochondrial stress are elevated above wild-type levels when UPR$^{mt}$ is blocked by *atfs-1* RNAi. These data, paired with the fact that compromising HOE-1 nuclear export triggers UPR$^{mt}$, lead us to hypothesize that upon mitochondrial stress, nuclear HOE-1 levels are elevated. This upregulation of nuclear HOE-1 elevates 3'-tRNA processing thereby triggering a signaling cascade that results in elevated nuclear ATFS-1 and DVE-1 and subsequent UPR$^{mt}$ induction. Activated UPR$^{mt}$ then negatively regulates HOE-1 nuclear levels thus providing a feedback mechanism to tightly control mitochondrial stress response. UPR$^{mt}$-negative regulation of HOE-1 is further supported by our data showing that constitutive activation of UPR$^{mt}$ by *atfs-1(et15)* is sufficient to reduce nuclear HOE-1 levels in the absence of mitochondrial stress. How it is that mitochondrial stress activates HOE-1 is still unknown. Multiple mitochondrial derived small molecules have been reported to communicate mitochondrial status including reactive oxygen species (ROS), NAD+, and acetyl-CoA (*Baker et al., 2012*; *Mouchiroud et al., 2013*; *Ramachandran et al., 2019*; *Tjahjono et al., 2020*; *Zhu et al., 2020*) We look forward to further investigating whether these, or other molecules, are involved in HOE-1 regulation.

In humans, mutations in the ortholog of HOE-1, ELAC2, are associated with both hypertrophic cardiomyopathy (*Haack et al., 2013*; *Shinwari et al., 2017*; *Saoura et al., 2019*) and prostate cancer (*Tavtigian et al., 2001*; *Korver et al., 2003*; *Noda et al., 2006*). Historically, it has been suggested that mutations in ELAC2 cause disease because of a loss of mature tRNA production. Our works suggests an intriguing alternative whereby ELAC2 mutations lead to altered tRNA processing that triggers aberrant stress response signaling resulting in disease state. Our system provides a convenient opportunity to interrogate these disease causing variants.

Taken together, our findings provide a novel mechanism—involving the tRNA processing enzyme HOE-1—by which mitochondrial stress is transduced to activate UPR$^{mt}$ thus providing important insight into the regulation of mitochondrial stress response.

## Methods

### Worm maintenance
Worms were grown on nematode growth media (NGM) seeded with OP50 *E. coli* bacteria and maintained at 20 °C.

### Mutants and transgenic lines
A complete list of *C. elegans* strains used can be found in *Supplementary file 1*. All new mutant and transgenic strains generated via CRISPR/*Cas9* for this study were confirmed by Sanger sequencing.

### CRISPR/Cas9
CRISPR was conducted as previously described (Dokshin et al. Genetics 2018; Paix et al. Genetics 2015) using Alt-R S.p. Cas9 Nuclease V3 (IDT #1081058) and tracrRNA (IDT #1072532). A complete list of crRNA and repair template sequences purchased from IDT can be found in *Supplementary file 2*.

## Genetic crosses

Strains resulting from genetic crosses were generated by crossing ~20 heterozygous males of a given strain to 5–8 L4 hermaphrodites of another strain (heterozygous males were generated by first crossing L4 hermaphrodites of that strain to N2 males). F1, L4 hermaphrodites were then cloned out and allowed to have self-progeny. F2 progeny were cloned out and once they had progeny were genotyped or screened (if fluorescent marker) for presence of alleles of interest. All genotyping primers were purchased from IDT and can be found in *Supplementary file 2*.

## Fluorescence microscopy

All whole animal imaging was done using Zeiss Axio Zoom V16 stereo zoom microscope. For all whole animal imaging, worms were immobilized on 2% agar pads on microscope slides in ~1 µl of 100 mM levamisole (ThermoFisher #AC187870100) and then coverslip applied.

## Fluorescence image analysis

For whole animal fluorescence intensity quantification, total pixels (determined by tracing individual animals and summing the total number of pixels within the bounds of the trace) and pixel fluorescence intensity (pixel fluorescence intensity on 1–255 scale) were quantified using imageJ and mean fluorescence intensity for each worm was calculated (sum total of fluorescence intensity divided by total number of pixels within bounds of the trace). For DVE-1::GFP image analysis (*Figure 5E&F*), brightness threshold was set to 25 in imageJ and then the number of gut cell nuclei that were saturated at this threshold were counted. For *Figure 8A&B* and *Figure 8E&F*, and *Figure 8—figure supplement 2A&B*, mean fluorescence intensity was calculated within the bounds of gut cell nuclei and outside of the bounds of gut cell nuclei and then graphed as the ratio fluorescence intensity of nuclear to extranuclear signal.

## RNAi

RNAi by feeding was conducted as previously described (Gitschlag et al. Cell Met. 2016). Briefly, RNAi clones were grown overnight from single colony in 2 ml liquid culture of LB supplemented with 50 µg/ml ampicillin. To make 16 RNAi plates, 50 ml of LB supplemented with 50 µg/ml ampicillin was inoculated with 500 µl of overnight culture and then incubated while shaking at 37 °C for 4–5 hours (to an $OD_{550-600}$ of about 0.8). Cultures were then induced by adding 50 ml additional LB supplemented with 50 µg/ml ampicillin and 4 mM IPTG and then continued incubating while shaking at 37 °C for 4 hours. Following incubation, bacteria were pelleted by centrifugation at 3900 rpm for 6 min. Supernatant was decanted and pellets were gently resuspended in 4 ml of LB supplemented with 8 mM IPTG. 250 µl of resuspension was seeded onto standard NGM plates containing 1 mM IPTG. Plates were left to dry overnight and then used within 1 week. Bacterial RNAi feeder strains were all from Ahringer RNAi Feeding Library, grown from single colony and identity confirmed by Sanger sequencing. *atfs-1* (ZC376.7), *cco-1* (F26E4.9), *hoe-1* (E04A4.4), *hpo-31* (F55B12.4), *popl-1* (C05D11.9), *rpc-1* (C42D4.8), *rtcb-1* (F16A11.2), *spg-7* (Y47G6A.10), *xpo-3* (C49H3.10).

## Quantification of gene expression

cDNA was synthesized using Maxima H Minus First Strand cDNA Synthesis Kit, with dsDNase (ThermoFisher #K1682) according to manufacturer's directions. Lysates for cDNA synthesis were made by transferring 10, day 2 adult worms to 10 µl of lysis buffer supplemented with 20 mg/ml proteinase K and incubating at 65 °C for 10 min, 85 °C for 1 min and 4 °C for 2 min. Quantification of gene expression was performed using droplet digital PCR (ddPCR) with Bio-Rad QX200 ddPCR EvaGreen Supermix (Bio-Rad #1864034). Primers used for ddPCR can be found in *Supplementary file 2*.

## TMRE staining

A total of 500 µl of 1 mM TMRE (ThermoFisher #T669) solution in M9 buffer (prepared from a stock TMRE solution of 0.5 M in DMSO) was supplemented on top of standard NGM plates pre-seeded with 200 ul lawn of OP50 and allowed to dry overnight in the dark. The following day young L4 animals were transferred to TMRE plates and incubated on TMRE for 16 hr. After 16 hr, animals were transferred from TMRE plates to seeded standard NGM plates for 1 hr to remove any non-specific

TMRE signal from cuticle and intestinal lumen. Animals were then imaged via confocal microscopy as described below.

## Confocal fluorescence imaging

Worms were grown at 20 °C and age-synchronized by timed egg-lays on NGM plates seeded with OP50 or HT115 bacteria for RNAi experiments. Before imaging, worms were immobilized with 3 µl 0.05 µm Polybead microsphere suspension (Polysciences) on a 10% agarose pad with a coverslip (1). Images were taken in the mid- or posterior intestine using a Nikon Ti2 with CSU-W1 spinning disk and Plan-Apochromat 100 X/1.49 NA objective. HOE-1::GFP was imaged by 488 nm laser excitation and ET525/36 m emission filter. 2 X integration was applied (Nikon Elements) to increase signal strength. TMRE and ATFS-1::mCherry were imaged with 561 nm laser excitation and ET605/52 M emission filter.

Image processing and analysis was performed with Nikon Elements software. Raw images were subjected to deconvolution and rolling ball background subtraction. Mitochondrial networks were segmented using the TMRE signal after excluding dye aggregates via Bright Spot Detection. To objectively set threshold parameters across groups with different TMRE intensity levels, the low threshold for segmentation was calculated based on a linear correlation with mean TMRE intensity within each group, $y = 0.6411*x + 89.71$ (x = mean TMRE intensity and constants derived from an initial manual validation). Regions of interest (ROIs) were manually drawn to encompass a single intestinal cell, and nuclei were identified and segmented manually using brightfield images. Mean intensities were measured within the resulting masks.

To detect localization of HOE-1::GFP in mitochondria, images of TMRE-stained intestinal cells of control and ΔMTS worms were collected and blinded. Mitochondria were segmented by TMRE signals as above. For each cell, one representative line scan was drawn manually across the mitochondrial short axis.

## Western blot

Fifty adult worms were transferred into a tube containing 20 µl of M9 Buffer. Then, 20 µl of 2 x Laemmli Buffer (BioRad #161–0737) supplemented with 2-mercaptoethanol (i.e. βME) was added to worm suspension and gently pipetted up and down 5 times to mix. Worms were lysed at 95 °C for 10 min in thermocycler followed by ramp down to room temperature (25 °C). Lysates were then pipetted up and down 10 times to complete disrupt and homogenize suspension. Samples were briefly centrifuged to pellet any worm debris. 20 µl of lysate supernatant was loaded onto precast Mini-PROTEAN TGX Stain-Free Gel (BioRad #4568045). Gel was run for 30 min at 100 V and then an additional 40–45 min at 130 V in 1 x Tris/Glycine/SDS Running Buffer (BioRad #1610732). Following electrophoresis gel was activated and imaged for total protein. Gel was equilibrated in Trans-Blot Turbo Transfer Buffer (BioRad #10026938) and transferred to activated and equilibrated Trans-Blot Turbo LF PVDF Membrane (BioRad #10026934) for 7 min at 2.5 A/25 V on Trans-Blot Turbo Transfer System. Following transfer, stain-free membrane was imaged for total protein. Membrane was then blocked in 5% milk in TBST for 2 hr rocking at room temperature. Following blocking, membrane was incubated in primary antibody overnight rocking at 4 °C. Mouse monoclonal anti-β-actin (Santa Cruz Biotechnology #sc-47778) or mouse monoclonal anti-GFP (#sc-9996) were used at a dilution of 1:2,500 in 5% milk in TBST. The following day the membrane was washed three times for 5 min each with TBST and then incubated with HRP-conjugated goat anti-mouse antibody (sc-2005) at 1:2000 in 5% milk in TBST for 2 hours at room temperature. Membrane was again washed three times for 5 min each with TBST. Membranes were then incubated for 5 min in Clarity Western ECL Substrate (BioRad #1705060) and immediately imaged on a BioRad ChemiDoc MP imager. Band intensity was quantified using imageJ.

## Statistical analysis

Experiment-specific details regarding sample size and statistical test used can be found in the corresponding Figure Legends. Significant p-values under 0.05 are denoted on all graphs and p-values above 0.05 are considered non-significant (ns). All statistical analysis was performed in GraphPad Prism 9. All data points for each experiment are included (no outlier exclusion was performed). For all whole animal fluorescence analysis, a sample size of 24 animals was generally used, each animal considered a biological replicate. Statistical analysis of high resolution fluorescence confocal imaging (HOE-1::GFP, ATFS-1::mCherry, and TMRE) was conducted on sample sizes between 60 and 80

animals of which animals were collected and imaged on three independent days, each animal considered a biological replicate. For western blot analysis, four independent samples were used for each condition, each sample (containing 50 worms each) is considered a biological replicate. For ddPCR analysis, a sample size of 4 was used for each condition, each sample (containing 10 worms each) is considered a biological replicate, each biological replicate was run in technical duplicate of which the average value was used for analysis.

## Acknowledgements

We thank Lantana K Grub and Cassidy A Johnson for their valuable feedback on the manuscript. We thank WormBase for invaluable tools and information used to plan and execute the research described. Worm strain itSi001 was graciously shared with us by Sasha de Henau. Some strains were provided by the CGC, which is funded by NIH Office of Research Infrastructure Programs (P40 OD010440). This work was generously supported by R01 GM123260 (MRP), R35 GM145378 (MRP), R00 AG052666 (KB), and by the support provided to JPH by the Training Program in Environmental Toxicology (T32ES007028). Some confocal microscopy imaging was performed through the Vanderbilt Cell Imaging Shared Resource (supported by NIH grants CA68485, DK20593, DK58404, DK59637 and EY08126). Droplet Digital PCR to quantify transcript levels was performed through the Vanderbilt University Medical Center's Immunogenomics, Microbial Genetics and Single Cell Technologies core.

## Additional information

### Funding

| Funder | Grant reference number | Author |
| --- | --- | --- |
| National Institute of General Medical Sciences | R01GM123260 | James P Held<br>Benjamin R Saunders<br>Maulik R Patel<br>Claudia V Pereira |
| National Institute on Aging | R00AG052666 | Gaomin Feng<br>Kristopher Burkewitz |
| National Institute of Environmental Health Sciences | T32ES007028 | James P Held |
| National Institute of General Medical Sciences | R35GM145378 | James P Held<br>Benjamin R Saunders<br>Maulik R Patel |

The funders had no role in study design, data collection and interpretation, or the decision to submit the work for publication.

### Author contributions

James P Held, Conceptualization, Formal analysis, Investigation, Methodology, Project administration, Validation, Visualization, Writing - original draft, Writing – review and editing; Gaomin Feng, Formal analysis, Investigation, Methodology, Validation, Writing – review and editing; Benjamin R Saunders, Investigation; Claudia V Pereira, Investigation, Writing – review and editing; Kristopher Burkewitz, Funding acquisition, Supervision, Writing – review and editing; Maulik R Patel, Conceptualization, Funding acquisition, Project administration, Resources, Supervision, Writing – review and editing

### Author ORCIDs

James P Held http://orcid.org/0000-0003-4322-2108
Maulik R Patel http://orcid.org/0000-0003-3749-0122

### Decision letter and Author response

Decision letter https://doi.org/10.7554/eLife.71634.sa1
Author response https://doi.org/10.7554/eLife.71634.sa2

## Additional files

### Supplementary files
- Transparent reporting form
- Supplementary file 1. C. elegans strains used in this study.
- Supplementary file 2. Oligonucleotides used in this study.

### Data availability
All data generated or analyzed during this study are included in the manuscript and supporting files. Source data files have been provided for Figures 5 and 8.

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
