## [Editor Report]

This manuscript reports a novel RNA-based cellular pathway that modulates mitochondrial UPR (UPRmt). It advances our understanding of the mitochondrial-to-nuclear communication mediated by a tRNA processing enzyme.

---

## [Decision Letter]

**Decision letter after peer review:**

Thank you for submitting your article "A tRNA processing enzyme is a central regulator of the mitochondrial unfolded protein response" for consideration by *eLife*. Your article has been reviewed by 3 peer reviewers, one of whom is a member of our Board of Reviewing Editors, and the evaluation has been overseen by David Ron as the Senior Editor. The reviewers have opted to remain anonymous.

Essential revisions:

The following points (especially the major experiments) should be addressed to strengthen the conclusion that HOE-1 plays a specific role in the activation of mtUPR.

Major experiments:

1. The authors use the transcriptional reporter hsp-6::gfp as a mtUPR reporter.

However, a fluorescent signal requires not only transcription from the hsp-6 promoter (the parameter of interest) but also translation of the derived mRNA. As HOE-1 is a tRNA processing enzyme whose inactivation may affect protein synthesis, qRT-PCR analysis (or some alternative analytical strategy) should be performed to quantify the effects of HOE-1 inhibition on the mtUPR transcription response, independent of the translation of a reporter.

Several groups have shown that inhibition of S6 kinase inhibits mtUPR activation. As HOE-1 is presumably required for protein synthesis, perhaps the mechanism is related? Does inhibition of other genes affecting tRNA levels also impair mtUPR or is it specific to HOE-1?

2. It seems that the HOE-1 protein with a mitochondrial targeting sequence is

transcribed from the same gene as HOE-1 without the MTS. And there are separate transcriptional start sites for each mRNA/protein. Considering the number of claims related to subcellular localization of HOE-1, the authors must determine if transcription from either site is altered during mitochondrial stress. During mitochondrial stress, does the ratio of HOE-1 transcript change? For example, is the hoe-1 variant transcript lacking the MTS increased?

3. There is an important caveat regarding the interpretation of the hoe-1(∆NES) strain which causes mtUPR activation: It remains unclear if nuclear accumulation is an event driving mtUPR activation or if the activation reflects a different feature of the ∆NES mutation.

The authors suggest that mtUPR induction in hoe-1(∆NES) is a result of increased 3'-tRNA processing. Whether 3'-tRNA processing is elevated in hoe-1(∆NES) should be tested more directly. Is it possible to determine the tRNA species that are elevated in hoe-1(∆NES) strain by sequencing? Or that the authors can express hoe-1(∆NES) that lacks the enzymatic activity and see whether it can still activate mtUPR.

4. There is a concern In regards to the finding that hoe-1(ΔNES) mutant is sufficient to induce the nuclear accumulation of the ATFS-1 and the subsequent up-regulation of the mtUPR reporter gene: the authors did not rule out the possibility that mitochondrial protein homeostasis was already disrupted in hoe-1(ΔNES) mutants so that the mtUPR was induced. Does HOE-1∆NES cause mitochondrial dysfunction which increases mtUPR activation? The authors only showed that mitochondrial membrane potential was not changed in hoe-1(ΔNES) mutants. More characterization of mitochondrial function in hoe-1(ΔNES) mutants is required, such as OCR and mitochondrial morphology. It seems that hoe-1(ΔNES) mutants are smaller than wild-type animals. Alternatively, the ∆NES mutation could be combined with the ∆MTS mutation.

5. The authors generate a beautiful ATFS-1::mCherry fusion protein and

demonstrate that it accumulates within nuclei during mitochondrial stress. Why is the overall level of ATFS-1 dramatically increased in hoe-1(ΔNES) mutants (Figure 4a)? This is not consistent with only two-fold up-regulation of atfs-1 transcript levels. Does hoe-1 inhibition affect translation/synthesis of ATFS-1::mCherry or nuclear accumulation of ATFS-1::mCherry? Or, DVE-1?

The authors also need to show the ATFS-1::GFP expression pattern in the nuo-6 mutants as a control.

6. Regarding the specific involvement of HOE-1 in the regulation of mtUPR, since

tRNA processing, the tRNA exporter xpo-3, as well as the RNase P complex popl-1, are all general regulators for protein synthesis. How to explain the specific involvement of these regulators only in the regulation of the mtUPR? The authors mentioned that HOE-1 homolog ELAC2 is not only required for tRNA maturation but also essential for the formation of tRNA fragments, snoRNAs, and miRNAs, are these non-coding RNAs account for the activation of the mtUPR?

It is also confusing that HOE-1(∆NLS) mutants suppressed the mtUPR induction in nuo-6 mutants, however, xpo-3 which functions in the same pathway as HOE-1 in terms of tRNA processing and export did not suppress the mtUPR induction in nuo-6 mutants in Figure 6i and 6j.

*Reviewer #2 (Recommendations for the authors):*

It was reported that the epigenetic regulation of the UPRmt is in parallel with the ATFS-1 pathway (PMID: 27133168, PMID: 27133166). Whether these epigenetic factors are required for the induction of UPRmt in hoe-1(ΔNES) mutants. Similarly, whether HOE-1(NLS) suppressed the epigenetic changes or the accumulation of epigenetic factors (PMID: 27133166, PMID: 32789178, PMID: 32934238 ) in response to mitochondrial stresses.

---

## [Author Response]

Essential revisions:The following points (especially the major experiments) should be addressed to strengthen the conclusion that HOE-1 plays a specific role in the activation of mtUPR.Major experiments:1. The authors use the transcriptional reporter hsp-6::gfp as a mtUPR reporter.However, a fluorescent signal requires not only transcription from the hsp-6 promoter (the parameter of interest) but also translation of the derived mRNA. As HOE-1 is a tRNA processing enzyme whose inactivation may affect protein synthesis, qRT-PCR analysis (or some alternative analytical strategy) should be performed to quantify the effects of HOE-1 inhibition on the mtUPR transcription response, independent of the translation of a reporter.

The reviewers raise an important point. To determine the effects of *hoe-1* inhibition on the UPR^mt^ transcriptional response independent of translation of the UPR^mt^ reporter (*hsp-6p::GFP*) we performed droplet digital PCR to quantify transcripts of genes upregulated upon UPR^mt^ activation (i.e. *hsp-6* and *cyp-14A4.1*) in a wildtype and *hoe-1(ΔNLS)* background in the absence and presence of mitochondrial stress (*control* and *spg-7* RNAi, respectively). We find that loss of nuclear HOE-1 results in attenuation of both *hsp-6* and *cyp-14A4.1* transcript levels in mitochondrial stress conditions (Figure 2I and Figure 2 —figure supplement 5A). This finding is consistent with the effect of loss of nuclear HOE-1 on UPR^mt^ reporter induction and further suggests that nuclear HOE-1 is directly involved in UPR^mt^ transcriptional response.

Several groups have shown that inhibition of S6 kinase inhibits mtUPR activation. As HOE-1 is presumably required for protein synthesis, perhaps the mechanism is related? Does inhibition of other genes affecting tRNA levels also impair mtUPR or is it specific to HOE-1?

The reviewers query whether inhibition of other genes affecting tRNA levels also impair UPR^mt^. To address this question we assessed UPR^mt^ reporter activation in mitochondrial stressed animals when other tRNA processing genes are knocked-down. These include RNA polymerase III subunit, *rpc-1* (Figure 6 —figure supplement 2C, 2D), RNAse P subunit, *popl-1* (Figure 6C, 6D), and tRNA ligase, *rtcb-1* (Figure 6 —figure supplement 3D, 3E). RNA polymerase III transcribes tRNAs, RNAse P processes 5’ ends of nascent tRNAs before they are processed at the 3’ end by HOE-1, and the tRNA ligase is involved in splicing of intron-containing tRNAs. Knockdown of *rpc-1* did not significantly impact *nuo-6(qm200)* induced UPR^mt^. Knock-down of *popl-1* and *rtcb-1* partially attenuate UPR^mt^ activation by *nuo-6(qm200).* These data suggest that the inhibition of UPR^mt^ is not specific to *hoe-1* loss-of-function and further strengthen the connection between tRNA biology and UPR^mt^.

Although not asked for directly, prompted by the reviewer suggestion, we also tested whether *rpc1* and *rtcb-1* knockdown impairs *hoe-1(ΔNES)*-induced UPR^mt^ (we had already reported in the original manuscript that *popl-1* RNAi suppresses *hoe-1(ΔNES)*-induced UPR^mt^). Like *popl-1* RNAi, *rpc-1* RNAi robustly attenuates *hoe-1(ΔNES)*-induced UPR^mt^ (Figure 6 —figure supplement 2A, 2B) further suggesting that limiting the availability of tRNA substrates for HOE-1 to act on suppresses *hoe-1(ΔNES)*-induced UPR^mt^ Knock-down of *rtcb-1* also mildly attenuates *hoe1(ΔNES)*-induced UPR^mt^ (Figure 6 —figure supplement 3A, 3B) suggesting that downstream rates of tRNA processing may also impact *hoe-1(ΔNES)*-induced UPR^mt^. These data provide further support for the role of tRNAs in inducing UPR^mt^.

The reviewers raise an interesting possibility that mTOR and HOE-1 mechanisms of UPR^mt^ induction may be related or intertwined. Given the broad involvement of mTOR signaling in cellular processes we would need to fully investigate any potential connection between these pathways (i.e., direct interaction) in future work.

2. It seems that the HOE-1 protein with a mitochondrial targeting sequence istranscribed from the same gene as HOE-1 without the MTS. And there are separate transcriptional start sites for each mRNA/protein. Considering the number of claims related to subcellular localization of HOE-1, the authors must determine if transcription from either site is altered during mitochondrial stress. During mitochondrial stress, does the ratio of HOE-1 transcript change? For example, is the hoe-1 variant transcript lacking the MTS increased?

We appreciate the reviewers’ suggestion to assess transcript dynamics of *hoe-1*. HOE-1 protein with and without a mitochondrial targeting sequence are indeed transcribed from the same gene locus. However, whether the two protein isoforms are independently transcribed is not clear. In fact, in human cell culture it has been shown that both mitochondrial and nuclear-targeted HOE1 are produced from the same transcript via alternative translation initiation (Rossmanith, PMID: 21559454). Thus, we first endeavored to determine the mode by which mitochondrial and nuclear HOE-1 are individually produced. We designed two sets of primers for measuring *hoe-1* transcript levels. One set that amplifies only transcripts containing the sequence encoding the mitochondrial targeting sequence and one set that amplifies all HOE-1 transcripts (i.e., sequence that is found in both mitochondrial and nuclear isoforms). If the two isoforms are a result of independent transcription, we would expect the number of mitochondrial specific transcripts to be lower than total transcript levels. However, using droplet digital PCR, we find that the number of transcripts that include a mitochondrial targeting sequence were nearly identical to the number of total *hoe1* transcripts (Figure 8 —figure supplement 1A). This finding suggests, that like in higher eukaryotes, HOE-1 is dual-targeted via differential translation of a single transcript.

Given the above finding, we next endeavored to determine if *hoe-1* transcript levels are altered upon mitochondrial stress. We find that hoe-1 transcript levels are mildly elevated under conditions of mitochondrial stress (i.e., *nuo-6(qm200)* worms) relative to wildtype when measured by ddPCR using both sets of aforementioned primers (Figure 8 —figure supplement 1B, 1C). These findings are consistent with our HOE-1 protein level analysis and support our finding that nuclear HOE-1 levels are elevated upon mitochondrial stress.

3. There is an important caveat regarding the interpretation of the hoe-1(∆NES) strain which causes mtUPR activation: It remains unclear if nuclear accumulation is an event driving mtUPR activation or if the activation reflects a different feature of the ∆NES mutation.The authors suggest that mtUPR induction in hoe-1(∆NES) is a result of increased 3'-tRNA processing. Whether 3'-tRNA processing is elevated in hoe-1(∆NES) should be tested more directly. Is it possible to determine the tRNA species that are elevated in hoe-1(∆NES) strain by sequencing? Or that the authors can express hoe-1(∆NES) that lacks the enzymatic activity and see whether it can still activate mtUPR.

The reviewers raise an important point regarding the functional nature of the *hoe-1(ΔNES)* mutant that we generated and used in the manuscript. To validate the function of the *hoe-1(ΔNES)* allele we conducted three complimentary experiments. First, as suggested, we created a catalyticallydead *hoe-1(ΔNES)* allele by introducing a point mutation (D624A) in *hoe-1* that ablates zinc binding. The endonuclease activity of HOE-1 is dependent upon zinc binding as it is a zinc phosphodiesterase. Homozygous *hoe-1(D624A+ΔNES)* animals have the same arrest phenotype as *hoe-1* null animals. Given that UPR^mt^ is not activated in *hoe-1(ΔNES)* animals until late in development we needed to be able to assess the impact of the D624A mutation later in development. To overcome this constraint we established *hoe-1(ΔNES)/hoe-1(D624A+ΔNES)* trans-heterozygous animals that expressed the UPR^mt^ reporter *hsp-6p::GFP*. These animals were able to grow to adulthood and thus we could assess impact on UPR^mt^ activation. *hoe1(ΔNES)/hoe-1(D624A+ΔNES)* trans-heterozygous animals had markedly diminished UPR^mt^ activation relative to homozygous *hoe-1(ΔNES)* animals (Figure 6 —figure supplement 1A, 1B) suggesting that the ability of *hoe-1(ΔNES)* to activate UPR^mt^ requires the RNA processing function of HOE-1.

Secondly, for *hoe-1(ΔNES)* to facilitate increased 3’-tRNA processing this would likely require there to be elevated nuclear HOE-1 levels in *hoe-1(ΔNES)* animals. To assess this we generated a C-terminally GFP-tagged *hoe-1(ΔNES)* allele *hoe-1(ΔNES::GFP)* and compared it’s subcellular expression to wildtype *hoe-1::GFP*. Based on high resolution imaging and its quantification, there is elevated HOE-1::GFP signal in nuclei of the *hoe-1(ΔNES)* background relative to wildtype (Figure 3 —figure supplement 1B, Figure 2 —figure supplement 4B, 4C). This finding is consistent with our hypothesis that there is increased 3’-tRNA processing in *hoe-1(ΔNES)* animals.

Third, and finally, if elevated nuclear HOE-1 levels are responsible for UPR^mt^ activation we reasoned that ablating HOE-1 nuclear localization in *hoe-1(ΔNES)* animals (*hoe-1(ΔNLS+ΔNES))* should inactivate *hoe-1(ΔNES)*-induced UPR^mt^. Indeed we found that compromising HOE-1 nuclear localization was sufficient to completely attenuate UPR^mt^ induced by *hoe-1(ΔNES)* (Figure 3 —figure supplement 3A, 3B). This finding strongly suggests that HOE-1 is required in the nucleus to activate UPR^mt^.

Combined, these experiments suggest that UPR^mt^ in *hoe-1(ΔNES)* animals is induced by increased 3’-tRNA processing that is a result of elevated nuclear levels of HOE-1.

4. There is a concern In regards to the finding that hoe-1(ΔNES) mutant is sufficient to induce the nuclear accumulation of the ATFS-1 and the subsequent up-regulation of the mtUPR reporter gene: the authors did not rule out the possibility that mitochondrial protein homeostasis was already disrupted in hoe-1(ΔNES) mutants so that the mtUPR was induced. Does HOE-1∆NES cause mitochondrial dysfunction which increases mtUPR activation? The authors only showed that mitochondrial membrane potential was not changed in hoe-1(ΔNES) mutants. More characterization of mitochondrial function in hoe-1(ΔNES) mutants is required, such as OCR and mitochondrial morphology. It seems that hoe-1(ΔNES) mutants are smaller than wild-type animals. Alternatively, the ∆NES mutation could be combined with the ∆MTS mutation.

We thank the reviewers for making this important suggestion to more thoroughly investigate the relationship between UPR^mt^ and mitochondrial function in *hoe-1(ΔNES)* animals. The experiments we conducted in response to these suggestions proved to be very informative. Compromised mitochondrial membrane potential has been shown to be the driving factor for UPR^mt^ activation as decreased membrane potential impairs mitochondrial import of proteins with weakly charged mitochondrial targeting sequences including ATFS-1 (Rolland *et al.,* PMID: 31412237, Shpilka *et al.,* PMID: 33473112). In the original draft of the manuscript, we had measured membrane potential in L4 stage animals and had not observed any differences between wildtype and *hoe1(ΔNES)* animals. However, as the UPR^mt^ is most robustly induced in 2-day old adult *hoe1(ΔNES)* animals, we reassessed membrane potential at this later stage. Furthermore, in collaboration with the Burkewitz Lab, this measurement was done using high resolution microscopy as opposed to whole animal imaging. We conducted TMRE staining on adult *hoe1(ΔNES)* and wildtype animals and found that mitochondrial membrane potential is significantly reduced in *hoe-1(ΔNES)* relative to wildtype (Figure 4A, 4B). Thus, these data are consistent with the reviewers’ surmise that there may be mitochondrial dysfunction in *hoe-1(ΔNES)* animals. Interestingly, *hoe-1(ΔNLS)* animals also show a similarly drastic decline in mitochondrial membrane potential (Figure 4A, 4B), despite the fact that UPR^mt^ is attenuated in this background. Thus, while there is a correlation between decreased membrane potential and UPR^mt^ induction in *hoe-1(ΔNES)* animals, it is difficult to infer causality between the two.

UPR^mt^ induction has been reported to cause a decrease in mitochondrial membrane potential. Therefore, we wondered whether UPR^mt^ causes decline in mitochondrial membrane potential in *hoe-1(ΔNES)* animals. To test for this possibility, we measured mitochondrial membrane potential using TMRE in *hoe-1(ΔNES)* on *atfs-1* RNAi. Loss of *atfs-1* did not rescue membrane potential in *hoe-1(ΔNES)* background. Based on these data, we conclude in the manuscript that *hoe-1(ΔNES)* directly causes a decrease in mitochondrial membrane potential independent of UPR^mt^.

In addition, we took the reviewers’ suggestion of creating a *hoe-1(ΔMTS+ΔNES)* mutant to address whether *hoe-1(ΔNES)* may be having a compromising effect directly in the mitochondria. If *hoe-1(ΔNES)* is causing UPR^mt^ by acting in the mitochondria, then impairing its mitochondrial localization should attenuate *hoe-1(ΔNES)*-induced UPR^mt^. If instead, as we hypothesized, *hoe1(ΔNES)* activates UPR^mt^ through its nuclear role, then compromising mitochondrial localization of HOE-1 should not attenuate *hoe-1(ΔNES)*-induced UPR^mt^. We found that *hoe-1(ΔMTS+ΔNES)* animals have higher UPR^mt^ activation than *hoe-1(ΔNES)* alone (Figure 3 —figure supplement 4A, 4B). This is consistent with HOE-1 activating UPR^mt^ via increased nuclear accumulation and rules out the possibility that mitochondrial localized HOE-1 induces UPR^mt^ in *hoe-1(ΔNES)* animals.

5. The authors generate a beautiful ATFS-1::mCherry fusion protein anddemonstrate that it accumulates within nuclei during mitochondrial stress. Why is the overall level of ATFS-1 dramatically increased in hoe-1(ΔNES) mutants (Figure 4a)? This is not consistent with only two-fold up-regulation of atfs-1 transcript levels. Does hoe-1 inhibition affect translation/synthesis of ATFS-1::mCherry or nuclear accumulation of ATFS-1::mCherry? Or, DVE-1?The authors also need to show the ATFS-1::GFP expression pattern in the nuo-6 mutants as a control.

To more thoroughly investigate ATFS-1 levels across backgrounds, with the help from the Burkewitz Lab, we conducted the ATFS-1::mCherry imaging experiments at high resolution using confocal microscopy as opposed to our original imaging which was done on a Nikon Ti-E Fluorescence Motorized DIC Polarization Microscope. In addition to wildtype and *hoe-1(ΔNES)* animals, we also imaged ATFS-1::mCherry in *nuo-6(qm200)* animals as a positive control, as suggested by the reviewers. High resolution microscopy of ATFS-1::mCherry confirmed our previous findings that nuclear ATFS-1 levels are elevated in *hoe-1(ΔNES)* (Figure 5A, 5B). Importantly, nuclear ATFS-1 levels were also elevated under mitochondrial stress (i.e. *nuo6(qm200)* animals) as expected (Nargund *et al.,* PMID: 22700657). We also quantified total cellular and extranuclear ATFS-1::mCherry fluorescence levels to address the reviewers’ question regarding the impact of *hoe-1(ΔNES)* on ATFS-1 translation/synthesis. *hoe-1(ΔNES)* animals do not exhibit elevated total or extranuclear ATFS-1::mCherry levels (Figure 5C and Figure 5 —figure supplement 1A). These data suggest that *hoe-1(ΔNES)* results in elevated nuclear localization but not increased ATFS-1 protein levels.

Similarly, to address the impact of *hoe-1(ΔNES)* on DVE-1 translation/synthesis level we conducted a western blot for DVE-1::GFP in a wildtype vs *hoe-1(ΔNES)* background. DVE-1 levels are not significantly different between wildtype and *hoe-1(ΔNES)* (Figure 5G, 5H, Figure 5 – source data 1) suggesting that *hoe-1(ΔNES)* triggers nuclear accumulation of DVE-1 as opposed to upregulating total DVE-1 protein levels.

6. Regarding the specific involvement of HOE-1 in the regulation of mtUPR, sincetRNA processing, the tRNA exporter xpo-3, as well as the RNase P complex popl-1, are all general regulators for protein synthesis. How to explain the specific involvement of these regulators only in the regulation of the mtUPR? The authors mentioned that HOE-1 homolog ELAC2 is not only required for tRNA maturation but also essential for the formation of tRNA fragments, snoRNAs, and miRNAs, are these non-coding RNAs account for the activation of the mtUPR? It is also confusing that HOE-1(∆NLS) mutants suppressed the mtUPR induction in nuo-6 mutants, however, xpo-3 which functions in the same pathway as HOE-1 in terms of tRNA processing and export did not suppress the mtUPR induction in nuo-6 mutants in Figure 6i and 6j.

We appreciate these reviewer comments. While indeed *xpo-3* and *popl-1* should be required for protein synthesis it is clear from our results that modulating their activity can specifically impact UPR^mt^. While surprising, these data support the idea that in addition to their role in protein synthesis more generally, tRNAs (or other putative HOE-1-processed RNAs) play a specific signaling role in modulating UPR^mt^. This idea explains how RNAi against essential tRNA processing machinery, while strong enough to compromise UPR^mt^ activation, is not strong enough to significantly impact protein synthesis. Indeed this reasoning is supported by the fact that animals on *xpo-3* and *popl-1* RNAi grow to adulthood.

Orthologs of HOE-1 have been reported to be capable of processing other RNA species. If those species are involved in UPR^mt^ regulation they would need to be transported by tRNA exportin (*xpo-3*)—while such xpo-3 dependent transport of non-tRNAs has not been shown to date, it is plausible. We have addressed this possibility in the discussion (manuscript page 17, lines 8-10) and look forward to identifying the causal RNA in future studies.

We agree that the differential impact of *xpo-3* RNAi on *hoe-1(ΔNES)*- and *nuo-6(qm200)*-induced UPR^mt^ is interesting. One reasonable hypothesis to explain this data is that while HOE-1 processed tRNAs play a role in activating UPR^mt^ in response to mitochondrial stress, ATFS-1 is also capable of activating UPR^mt^ directly. In contrast, HOE-1 processed tRNAs are presumably solely responsible for UPR^mt^ activation in *hoe-1(ΔNES)* animals and hence completely dependent on their exporter XPO-3. We hope to formally test this hypothesis once we identify the causal RNA species in the future.

Reviewer #2 (Recommendations for the authors):It was reported that the epigenetic regulation of the UPRmt is in parallel with the ATFS-1 pathway (PMID: 27133168, PMID: 27133166). Whether these epigenetic factors are required for the induction of UPRmt in hoe-1(ΔNES) mutants. Similarly, whether HOE-1(NLS) suppressed the epigenetic changes or the accumulation of epigenetic factors (PMID: 27133166, PMID: 32789178, PMID: 32934238 ) in response to mitochondrial stresses.

We thank the reviewer for this suggestion. We agree that the involvement of epigenetic regulation is a plausible and intriguing possibility. To thoroughly assess such involvement, we feel, is outside of the scope of the current manuscript. We look forward to addressing this in future studies.